# Improving Multi-modal Large Language Model through Boosting Vision Capabilities

## Abstract

We focus on improving the visual understanding capability for boosting the vision-language models. We propose **Arcana**, a multiModal language model, which introduces two crucial techniques. First, we present Multimodal LoRA (MM-LoRA), a module designed to enhance the decoder. Unlike traditional language-driven decoders, MM-LoRA consists of two parallel LoRAs – one for vision and one for language – each with its own parameters. This disentangled parameters design allows for more specialized learning in each modality and better integration of multimodal information. Second, we introduce the Query Ladder adapter (QLadder) to improve the visual encoder. QLadder employs a learnable "*ladder*" structure to deeply aggregates the intermediate representations from the frozen pretrained visual encoder (e.g., CLIP image encoder). This enables the model to learn new and informative visual features, as well as remaining the powerful capabilities of the pretrained visual encoder. These techniques collectively enhance Arcana's visual perception power, enabling it to leverage improved visual information for more accurate and contextually relevant outputs across various multimodal scenarios. Extensive experiments and ablation studies demonstrate the effectiveness and generalization capability of our Arcana.

## 1 Introduction

In recent years, multimodal large language models (MLLMs) Wang et al. (2023); Bai et al. (2023); Liu et al. (2024); Ye et al. (2023a) have made significant advancements. These models amalgamate image representations into large language models (LLMs) through an adaptor Touvron et al. (2023a); Zheng et al. (2024). Various methods Dai et al. (2024); Liu et al. (2024); Wang et al. (2023); Dong et al. (2024) leverage the capabilities of the powerful LLM to excel in various multimodal tasks.

While existing MLLMs showcase remarkable proficiency in multimodal tasks, they still face challenges in visual perception that is crucial for further tasks, such as reasoning or creation Chen et al. (2023a); Liu et al. (2023a). Fig. 1 (a) presents several examples that clearly highlight this issue. We observe deficiencies in current MLLMs regarding low-level visual perception, such as color and quantity, as well as high-level visual perception, such as small object detection and localization. Consequently, there is a pressing necessity to bolster the comprehension capabilities of existing MLLMs, specially for *vision*.

The insufficient visual perception capabilities of MLLMs can mainly be attributed to two factors: *decoder* and *visual encoder*. As depicted in Fig. 1(b), existing language driven decoder structures directly couple visual and language modalities. Such design not only disregards their unique characteristics but also may lead to information confusion, thus impairing the accurate understanding and processing of visual information. On the other hand, freezing visual encoder directly limits the ability to learn and represent visual information. Therefore, improving the visual perception requires rethinking the decoder design and optimizing the use of the visual encoder to better capture and process visual features.

As shown in Fig. 1(c), previous multimodal large language models (MLLMs) typically relied on CLIP as the visual encoder. However, research Tong et al. (2024); Xu et al. (2024) has revealed limitations in CLIP's ability to capture complex visual features. To address this, recent methods have incorporated self-supervised learning (SSL) pretrained models, such as DINOv2 Oquab et al. (2024), and fused their outputs with CLIP's features to enhance the visual encoder's representation

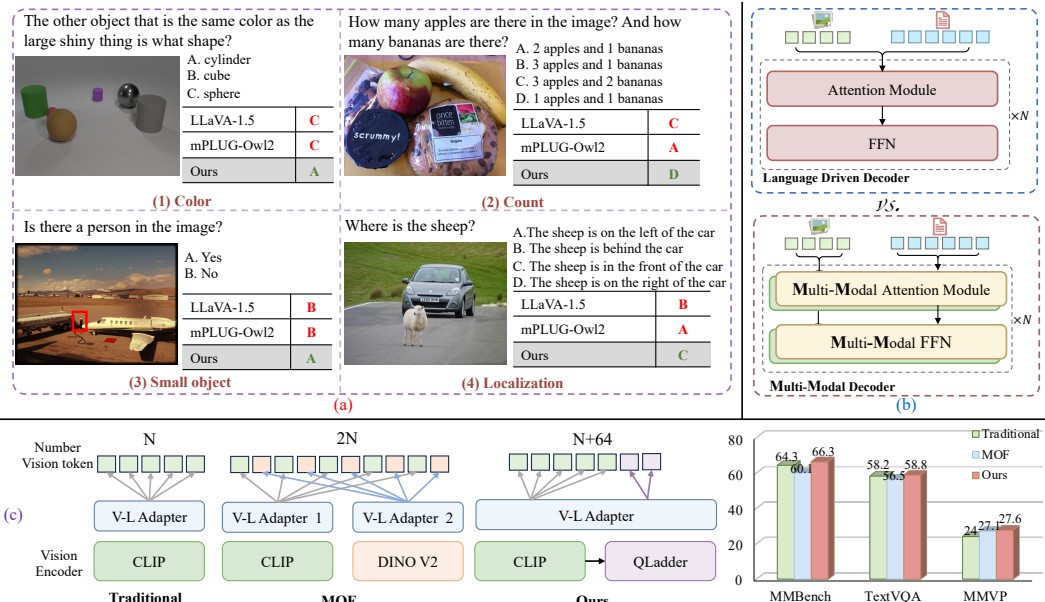

Figure 1: **(a)** Sampled some VQA examples involving color, quantity, small objects, and localization tasks, showcasing the importance of visual recognition capabilities for multimodal language models (MLLMs). **(b)** Contrasting Arcana's multimodal decoder with mainstream methods' language driven decoder. The language-driven decoder employs a language decoder (LLMs) directly to handle tokens from different modalities, which may lead to modality interference and performance degradation. In contrast, the multimodal decoder independently processes different token types to avoid modality interference. **(c)** illustrates the structures of different visual encoders and the resulting number of visual tokens obtained. The bar chart displays the model's performance across various architectures.

capacity. While this fusion approach improves feature representation, it also introduces significant computational overhead. The use of two visual encoders doubles the number of visual tokens, leading to a substantial increase in training costs, particularly when handling large-scale datasets and complex models.

Toward this end, we propose a new multimodal large language model **Arcana** that aims to enhance visual perception capabilities from both visual encoder and decoder. Specifically, we design a multimodal LoRA (MM-LoRA) to construct a multimodal decoder as show in Fig. 1(b). This decoder provides independent learning spaces for each modality, ensuring the decoupling of different modalities, avoiding information confusion, and preserving the uniqueness of each modality. Additionally, we propose a novel design, the Query Ladder Adapter (QLadder), as shown in Fig. 1(c). Unlike methods that significantly increase the number of visual tokens, our approach introduces only a small set of visual tokens (set to $64$, where $64 << N$). Despite the limited number of tokens, QLadder effectively enhances the model's ability to learn and represent visual information by progressively refining and integrating visual features through its *"ladder"* structure. Notably, even with the introduction of only a small number of visual tokens, QLadder achieves performance comparable to DINOv2-based MOF Tong et al. (2024) methods on the MMVP benchmark, which demands strong visual representations. Furthermore, our approach demonstrates performance improvements on traditional multimodal benchmarks, such as MMbench Liu et al. (2023b) and TextVQA Singh et al. (2019), highlighting its versatility and effectiveness across various tasks.

Finally, we conducted an extensive series of experiments to thoroughly evaluate the performance and effectiveness of Arcana. These experiments were designed to assess various aspects, including the robustness of MM-LoRA and QLadder across different benchmarks, its ability to generalize in diverse scenarios, and its performance in comparison to state-of-the-art methods.

## 2 RELATED WORK

**Multi-Modal Large Language Models.** Fueled by the tremendous success of large language models (LLMs) Achiam et al. (2023); Touvron et al. (2023a); Jiang et al. (2023), there is growing interest in developing end-to-end multi-modal large language models (MLLMs) Dai et al. (2024); Ye et al.

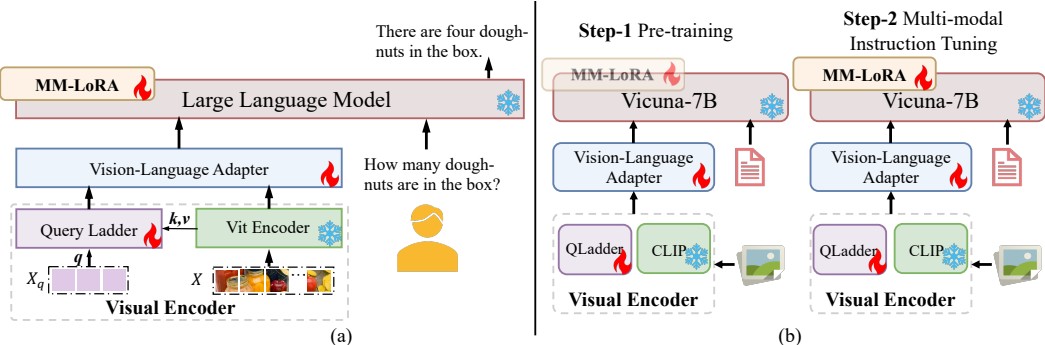

Figure 2: **(a)** The architecture of the Arcana. **(b)** The training pipeline of Arcana. MM-LoRA is optional during the pre-training phase.

(2023a); Dong et al. (2024). These models aim to enhance the visual perceptual capabilities of LLMs by integrating additional modalities, allowing for unified handling of multi-modal tasks. Currently, there are three primary approaches to building Multi-Modal foundational models, each demonstrating strong potential for zero-shot generalization in the visual-language domain.

The first approach, exemplified by Flamingo Alayrac et al. (2022), uses cross-attention to align visual models with large language models across modalities. The second approach, used by models like PaLM-E Driess et al. (2023), directly integrates extracted visual features into a pre-trained PaLM Anil et al. (2023) model via a linear layer, achieving robust performance. This method is widely adopted by mainstream models such as LLaVA Liu et al. (2024), CogVLM Wang et al. (2023) and Internlm-Xcomposer Zhang et al. (2023) but incurs high inference costs due to the lengthy visual tokens. To address this, the third approach, inspired by DETR Meng et al. (2021); Zhu et al. (2020) and represented by BLIP-2 Li et al. (2022), employs a Q-former to effectively reduce the sequence length of visual features. Similar designs are used by mPLUG-OWL2 Ye et al. (2023a), and MiniGPT-4 Zhu et al. (2023). However, these methods Anil et al. (2023); Bai et al. (2023); Chen et al. (2023a) couple visual and language modalities in the same space using language-guided decoders, overlooking the uniqueness of different modalities. This oversight may result in interference between modalities, potentially affecting performance. To this end, we employ MM-LoRA to implement a multimodal decoder, aiming to mitigate the impact of modality interference on the model.

**Improve visual perception for MLLMs.** Currently, MLLMs are the most popular approach in VL community Alayrac et al. (2022); Li et al. (2022), and enhancing their visual recognition capabilities has become a prominent research trend. Integrating visual features into large language models (LLMs) via a linear layer has become the mainstream approach Liu et al. (2024); Wang et al. (2023). However, this approach often relies on frozen vision encoders to provide visual features, which limits the visual recognition capabilities of multimodal large language models (MLLMs). To address this issue, existing methods enhance visual recognition in two ways. The first method Luo et al. (2024); Tong et al. (2024); Xu et al. (2024) introduces new high-resolution vision encoders, significantly improving visual recognition by increasing the number of visual tokens. For example, LLaVA-HR Luo et al. (2024) achieves this by incorporating ConvNeXt Liu et al. (2022) to handle high-resolution images. However, these methods significantly increases the number of visual tokens. Therefore, we propose QLadder, which can significantly enhance the model's visual perception capability with the introduction of a small number of visual tokens. The second method Wang et al. (2023); Dong et al. (2024); Ye et al. (2023a) expands the learning space for visual tokens within the large language model to accelerate visual-language alignment, thereby enhancing visual perception. For instance, Internlm-Xcomposer2 Dong et al. (2024) introduces Partial-LoRA, adding a LoRA to visual tokens to strengthen their representation. However, experiments with MM-LoRA have shown that directly increasing the learning space for visual tokens in the decoder does not improve the model's performance.

## 3 METHOD

### 3.1 OVERVIEW

We propose a new model, named Arcana as shown in Fig 2, designed to enhance visual perception in multimodal language models. Like most existing models Liu et al. (2024); Chen et al. (2023a), it

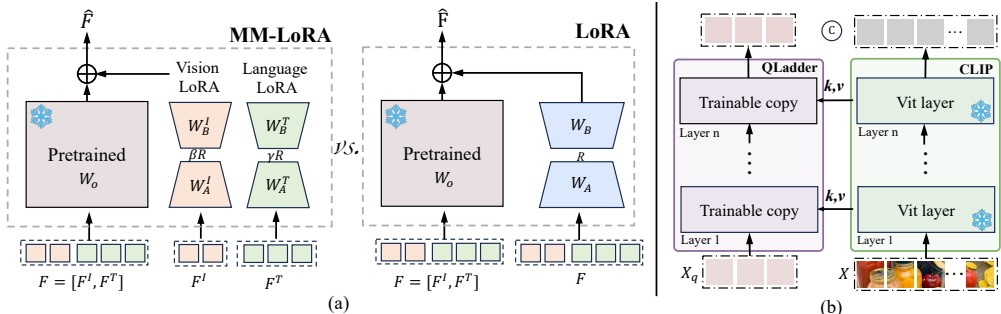

(a)

(b)

Figure 3: **(a)** The farmework of MM-LoRA *vs.* LoRA. MM-LoRA introduces two new hyperparameters, $\beta$ and $\gamma$, to control the ranks of the visual and language LoRAs, respectively. Notably, we set $\beta + \gamma = 1$ to ensure that MM-LoRA has the same number of parameters as LoRA. **(b)** The architecture of the visual encoder includes the QLadder adapter and CLIP. The QLadder adapter consists of cross-attention and FFN layers, with weights initialized from those of CLIP.

includes a visual encoder, a vision-language adapter, and a large language model. The key difference is that we use MM-LoRA to implement a multimodal decoder. Unlike traditional fine-tuning where visual and language modalities share LoRA parameters, MM-Lora assigns different LoRA parameters to each modality. Additionally, we introduce QLadder in the visual encoder, which significantly enhances the model's ability to learn and represent visual information with the introduction of a small number of visual tokens. We first briefly introduce Arcana's architecture in Section 3.2. Additionally, in Section 3.3, we detail MM-LoRA, which aims to decouple the learning spaces of different modalities to achieve a multimodal decoder. Lastly, we introduce the training paradigm of Arcana in Section 3.4.

## 3.2 ARCHITECTURE

Our approach Arcana (illustrated in Fig. 2(a) consists of three main components: visual encoder, vision-language adapter and large language model. Each component is described in the following.

**Visual Encoder.** The primary objective is to extract visual features from the image. The encoder learned with language-supervision, e.g., the CLIP Radford et al. (2021) visual model, is widely adopted. The CLIP encoder is often fixed during fine-tuning, e.g., in LLaVA, for keeping the representation capability of the original CLIP encoder. We propose to improve the visual encoder through a query ladder adaptor (QLadder) from the fine-tuning data that may contain new visual semantics. The structure is shown in Fig. 3(b). This adapter enhances the visual feature representation of the visual encoder by adding a small number of query visual tokens while retaining the pretrained image encoder. It improves Arcana's visual perception capability.

We extract visual features $\mathbf{F}_c \in \mathbb{R}^{N_I \times C_v}$ through the CLIP encoder, where $C_v$ represents the channel of visual feature, $N_I$ indicates the number of image patch. A set of learnable vectors $\mathbf{x}_q$ is fed into QLadder to acquire additional visual features $\mathbf{F}_q \in \mathbb{R}^{N_q \times C_v}$, where $N_q << N_I$. The two kinds of visual features are concatenated: $\mathbf{F}_v = \text{concat}(\mathbf{F}_c, \mathbf{F}_q)$. As shown in Fig 2(b), QLadder comprises multiple layers composed of cross-attention and feed-forward networks (FFNs).

**Vision-Language Adapter.** We map the output of visual encoder to the same space as the language embedding space through an Vision-Language adapter. The adapter g consists of two MLP layers. The output visual features are denoted as $\mathbf{F}^I = g(\mathbf{F}_v)$.

**Large Language Model.** For multimodal tasks Goyal et al. (2017); Hudson & Manning (2019), leveraging pre-trained large language models (LLMs) can provide valuable linguistic priors. Through multimodal instruction tuning, LLMs learn to comprehend visual features within images, enabling comprehensive understanding and processing of multimodal data. Typically, this process is accomplished through full fine-tuning or LoRA Hu et al. (2021). However, these methods overlook the unique characteristics of modalities, leading to modality confusion. This not only damages MLLMs' accurate understanding and processing of visual information but also affects natural language understanding. Therefore, a multimodal decoder that provides separate learning spaces for each modality is a better choice for MLLMs.

## 3.3 MULTIMODAL LORA

To implement a multimodal decoder based on a large language model, we propose a multimodal LoRA. This approach projects visual and language features into separate semantic spaces to decouple their representations, thereby avoiding modality interference. This allows Arcana to retain the unique characteristics of each modality, enhancing its visual perception without compromising natural language understanding. Next, we detail the MM-LoRA process.

MM-LoRA, as illustrated in Fig. 3, consists of visual LoRA and language LoRA. In comparison to LoRA, we introduces two parameters, $\beta$ and $\gamma$, to control the rank size of $(R)$ visual LoRA and language LoRA. It's worth noting that $\beta + \gamma = 1$ to ensure that no additional parameters are introduced compared to LoRA. Specifically, given a sequence of visual-language features $F \in \mathbb{R}^{(N_v + N_t) \times C}$ and a multimodal mask $M \in \{0, 1\}^{(N_v + N_t)}$, where $C$ represents the hidden dimension in LLMs, $N_v$ and $N_t$ indicates the number of visual and language tokens, respectively. We define a modality separation function $\Theta$ to separate the tokens of different modalities within $F$.

$$\Theta(F, M, m) = F \odot (M == m), \tag{1}$$

where $m \in \{0, 1\}$ is used to select between visual tokens ($m = 0$) and language tokens ($m = 1$). Therefore, based on multimodal mask $M$, we can obtain $F^I$ and $F^T$.

$$F^I = \Theta(F, M, 0) \qquad F^T = \Theta(F, M, 1) \tag{2}$$

Then, $F^I$ and $F^T$ are separately inputted into the visual part and language part of MM-LoRA. In Visual LoRA, the weights are denoted as $W_A^I \in \mathbb{R}^{C \times \beta R}$ and $W_B^I \in \mathbb{R}^{\beta R \times C}$, while in Language LoRA, the weights are denoted as $W_A^T \in \mathbb{R}^{C \times \gamma R}$ and $W_B^T \in \mathbb{R}^{\gamma R \times C}$.

Similarly to LoRA Hu et al. (2021), $F$ is inserted into the LLM layer to obtain $\hat{F}$. Finally, the output results of MM-LoRA are added to the output of LLM according to the mask $M_I$.

$$\hat{F} = W_o \times F$$
$$\Theta(\hat{F}, M, 0)+ = W_B^I \times W_A^I \times F^I \qquad \Theta(\hat{F}, M, 1)+ = W_B^T \times W_A^T \times F^T \tag{3}$$

In Arcana, MM-LoRA is applied to all linear layers of the large language model, thereby achieving an optimal multimodal decoder.

## 3.4 TRAINING PARADIGM

Following prior work Liu et al. (2024); Wang et al. (2023), we adopt a two-stage approach involving pretraining and multimodal instruction fine-tuning to train Arcana, as illustrated in Fig. 2(b). The purpose of the pretraining stage is to align the visual encoder with the language model, while multimodal instruction fine-tuning aims to adapt the model better to specific tasks through fine-tuning. We found that freezing the visual encoder limits the MLLM's ability to capture complex visual information, such as scene text and visual knowledge. To address this issue, we introduce Qladder and enable it to be trained in both the pretraining and instruction fine-tuning stages. This strategy allows the model to more effectively capture both low-level and high-level semantic visual information. Additionally, we introduce MM-LoRA fine-tuning as an alternative to full fine-tuning and LoRA fine-tuning, enabling a multimodal decoder that minimizes modality interference. Specifically, in the pretraining stage, we train Qladder and the vision-language adapter, while in the instruction fine-tuning stage, we train Qladder, the vision-language adapter, and MM-LoRA. Furthermore, to ensure the linguistic capabilities of Arcana, we employ joint training, adjusting the entire model during instruction fine-tuning, integrating textual and multimodal instructions.

## 4 EXPERIMENTS

### 4.1 IMPLEMENTATION DETAILS

**Model.** In the visual encoder, we utilize the CLIP-L Radford et al. (2021) model with an input resolution of 336 and a patch size of $14 \times 14$. Furthermore, the QLadder adapter adopts the same structure as CLIP-L, replacing self-attention with cross-attention. Notably, QLadder utilizes

Table 1: Performance on six General Visual Question Answering benchmarks. Specialist models, indicated in gray, are fine-tuned on each individual dataset. The red and blue colors respectively represent the optimal and suboptimal results on each benchmark. ∗ indicates that MM-LoRA is trained during the pretrain stage.

| Type | Model | LLM | In-domain VQA Tasks | | | Zero-shot VQA Tasks | | |
|------|-------|-----|------|-------|-----|--------|----------|------|
| | | | VQAv2 | OKVQA | GQA | TextVQA | ScienceQA | Ai2d |
| Generalists | BLIP2 Li et al. (2022) | Flan-T5 | 65.0 | 45.9 | 41.0 | 42.5 | 61.0 | - |
| | InstructBLIP Dai et al. (2024) | Vicuna (7B) | - | - | 49.2 | 50.1 | 60.5 | 40.6 |
| | InstructBLIP Dai et al. (2024) | Vicuna (13B) | - | - | 49.5 | 50.7 | 63.1 | - |
| | Shikra Chen et al. (2023a) | Vicuna (7B) | 77.4 | 47.2 | - | - | - | - |
| | IDEFICS-Instruct Laurençon et al. (2024) | LLaMA (65B) | 37.4 | 36.9 | - | 28.3 | 61.8 | 54.8 |
| | LLaVA-v1.5 Liu et al. (2023a) | Vicuna (7B) | 78.5 | - | **62.0** | 58.2 | 66.8 | 55.5 |
| | Qwen-VL-Chat Bai et al. (2023) | Qwen (7B) | 78.2 | 56.6 | 57.5 | **61.5** | 68.2 | - |
| | mPLUG-Owl2 Ye et al. (2023a) | LLaMA (7B) | **79.4** | 57.7 | 56.1 | 58.2 | 68.7 | 55.7 |
| | Arcana | Vicuna (7B) | 79.2 | **57.9** | 61.6 | **59.5** | **71.2** | **56.8** |
| | Arcana* | Vicuna (7B) | **79.5** | **58.9** | **61.8** | 58.7 | 69.5 | **56.9** |
| Specialists | GIT2 Wang et al. (2022) | - | 81.7 | - | - | 59.8 | - | - |
| | PaLI-17B Chen et al. (2022) | - | 84.3 | 64.5 | - | 58.8 | - | - |

pre-trained CLIP weights as its initial weights. For the LLMs, we employ the pre-trained Vicuna-7B Chiang et al. (2023) model. The Vision-Language adapter comprises two layer MLP. MM-LoRA, used for fully supervised multimodal instruction tuning, consists of a visual LoRA with a rank of $\beta \times R$ and a language LoRA with a rank of $\gamma \times R$.

**Data Sets.** During pre-training, we used approximately 1.2M image-text pairs from ShareGPT4V Chen et al. (2023b). In the multimodal instruction tuning stage, we utilize six types of supervised data totaling 934k, namely: (1) text-only instruction data (ShareGPT ShareGPT (2023)); (2) vision question-answering data (VQAv2 Goyal et al. (2017), GQA Hudson & Manning (2019), A-OKVQA Schwenk et al. (2022), OK-VQA Marino et al. (2019)); (3) OCR QA (OCRVQA Mishra et al. (2019), TextCaps Sidorov et al. (2020)); (4) Region-aware QA (RefCOCO Kazemzadeh et al. (2014); Mao et al. (2016), VG Krishna et al. (2017)); (5) multi-modal instruction data (LLaVA-instruct Liu et al. (2024)); and (6) image captions (VG-COCO Hao et al. (2024), shareGPT4V Chen et al. (2023b)). In the Ablation study, we only use the multimodal instruction data from LLaVA-v1.5.

**Training Setting.** During the pretraining step, we use language modeling loss with a batch size of 256 for 1 epoch. The learning rates are set to $1e-3$ for the vision-language adapter and $2e-5$ for Qladder. In the multimodal instruction tuning step, we integrated MM-LoRA into the LLM to create a multimodal decoder, thus preventing information interference between modalities. We set the learning rate for MM-LoRA to $1e-4$, and for both Qladder and the vision-language adapter, to $2e-5$. MM-LoRA is configured with a default rank $R$ of 256, $\beta$ set to 0.25, and $\gamma$ set to 0.75. All experiments are conducted on 8 NVIDIA A100 GPUs.

## 4.2 MAIN RESULTS

**General Visual Question Answering Benchmarks.** In Table 1, we compare with both SOTA MLLMs model on six General VQA benchmarks, including VQAv2 Goyal et al. (2017), OKVQA Schwenk et al. (2022), GQA Hudson & Manning (2019), TextVQA Singh et al. (2019), ScienceQA Lu et al. (2022) and Ai2d Kembhavi et al. (2016). We found that Arcana achieved competitive results on six VQA benchmarks. Notably, it achieved accuracies of 57.9 on OKVQA, 71.2 on ScienceQA, and 56.8 on Ai2d , surpassing most recently proposed MLLMs methods. Additionally, Arcana* with MM-LoRA used during the pre-training stage achieved better performance, indicating the importance of preserving the uniqueness of different modalities during pre-training. The superior performance on zero-shot VQA tasks particularly highlights strong generalization ability and potential across different domains of our model.

**Large Vision-Language Model Benchmarks.** Table 2 presents our comparative results on five different LVLM benchmarks: MMBench Liu et al. (2023b), MM-Vet Yu et al. (2023), SEED-Bench Li et al. (2023b), LLava$^W$ Liu et al. (2024), and POPE Li et al. (2023c). It is evident that Arcana achieves highly competitive performance across these benchmarks. Compared to mPLUG-

Table 2: Performance on five Large Vision-Language Models (LVLM) benchmarks.The red and blue colors respectively represent the optimal and suboptimal results on each benchmark. ∗ indicates that MM LoRA is trained during the pretrain stage.

| Method | Vision Encoder | Language Model | MME | MMBench | MM-Vet | SEED-Bench | LLaVA$^W$ | POPE |
|--------|----------------|----------------|-----|---------|--------|------------|-----------|------|
| BLIP-2 Li et al. (2022) | ViT-g (1.3B) | Vicuna (7B) | 1293.84 | - | 22.4 | 46.4 | 38.1 | 85.3 |
| MiniGPT-4 Zhu et al. (2023) | ViT-g (1.3B) | Vicuna (7B) | 581.67 | 23.0 | 22.1 | 42.8 | 45.1 | - |
| LLaVA Liu et al. (2024) | ViT-L (0.3B) | Vicuna (7B) | 502.82 | 36.2 | 28.1 | 33.5 | 63.0 | 80.2 |
| mPLUG-Owl Ye et al. (2023a) | ViT-L (0.3B) | LLaMA (7B) | 967.34 | 46.6 | - | 34.0 | - | - |
| InstructBLIP Dai et al. (2024) | ViT-g (1.3B) | Vicuna (7B) | 1212.82 | 36.0 | 26.2 | 53.4 | 60.9 | 78.9 |
| LLaMA-Adapter-v2 Gao et al. (2023) | ViT-L (0.3B) | LLaMA (7B) | 1328.40 | 39.5 | 31.4 | 32.7 | - | - |
| Otter Li et al. (2023a) | ViT-L (0.3B) | LLaMA (7B) | 1292.26 | 48.3 | 24.6 | 32.9 | - | - |
| Qwen-VL-Chat Bai et al. (2023) | ViT-G (1.9B) | Qwen (7B) | 1487.58 | 60.6 | - | 58.2 | - | - |
| LLaVA-v1.5 Liu et al. (2023a) | ViT-L (0.3B) | Vicuna (7B) | 1510.70 | 64.3 | 30.5 | 58.6 | 63.4 | 85.9 |
| mPLUG-Owl2 Ye et al. (2023b) | ViT-L (0.3B) | LLaMA (7B) | 1450.19 | 64.5 | 36.2 | 57.8 | - | 86.2 |
| **Arcana** | ViT-L (0.3B) | Vicuna (7B) | 1476.48 | 66.9 | 34.8 | 62.6 | 67.3 | 86.5 |
| **Arcana**∗ | ViT-L (0.3B) | Vicuna (7B) | 1520.93 | 67.4 | 34.4 | 63.2 | 72.7 | 87.1 |

OWL2 Ye et al. (2023b), Arcana scores 2.4 and 4.8 points higher on MMBench and SEED-Bench, respectively. Additionally, Arcana achieves a score of 86.5 on the hallucination evaluation dataset POPE, indicating significant advancements in visual recognition capabilities. These impressive results not only demonstrate its strong reasoning and multi-task generalization abilities but also clearly show that Arcana significantly outperforms others in these areas. Notably, we achieved this using a 0.3B visual encoder, with MM-LoRA and QLadder significantly enhancing the model's visual perception and generalization.

**Natural Language Understanding.** Although MLLMs excel in various multimodal downstream tasks, existing work Liu et al. (2024); Dong et al. (2024) often overlooks their natural language understanding capabilities. To address this, we also evaluated our model's language understanding performance on BIG-Bench Hard (BBH) Suzgun et al. (2023), AGIEval Zhong et al. (2023), and ARC Clark et al. (2018), as shown in Table 3. Compared to LLaMA-like Touvron et al. (2023a) language models, Arcana achieved competitive results across multiple benchmarks. This demonstrates that our model not only performs well in multimodal tasks but also excels in language understanding, further highlighting the superiority of our approach.

Table 3: **Performance on language benchmarks of our model** compared to LLaMA-2 0-shot for BBH, AGIEval, ARC.

| Method | BBH | AGIEval | ARC-c | ARC-e |
|--------|-----|---------|-------|-------|
| LLaMA-2 Touvron et al. (2023b) | 38.2 | 21.8 | 40.3 | 56.1 |
| WizardLM Xu et al. (2023) | 34.7 | 23.2 | 47.5 | 59.6 |
| LLaMA-2-Chat Touvron et al. (2023b) | 35.6 | 28.5 | 54.9 | 71.6 |
| Vicuna-v1.5 Chiang et al. (2023) | 41.2 | 21.2 | 56.6 | 72.8 |
| **Arcana** | **42.1** | **29.3** | **61.4** | **78.3** |

## 4.3 Ablation Study

To validate the effectiveness of QLadder and MM-LoRA, we designed a series of experiments. Additionally, to ensure fairness, we used only LLaVA-v1.5 Liu et al. (2023a) data for these experiments.

**Multimodal LoRA (MM-LoRA).** To validate the effectiveness of the multimodal decoder, we compared the performance of MM-LoRA and LoRA. Additionally, to investigate the importance of visual tokens and language tokens in the multimodal instruction tuning process within the decoder, we compared different ratios of $\beta$ and $\gamma$ parameters. In all experiments, the RANK of MM-LoRA and LoRA was set to 256. The results are shown in Table 4. It clearly indicate that MM-LoRA achieves optimal performance when $\beta = 0.25$ and $\gamma = 0.75$. When $\beta$ is set to 1, performance significantly drops, indicating that aligning language distribution using only visual tokens is challenging for

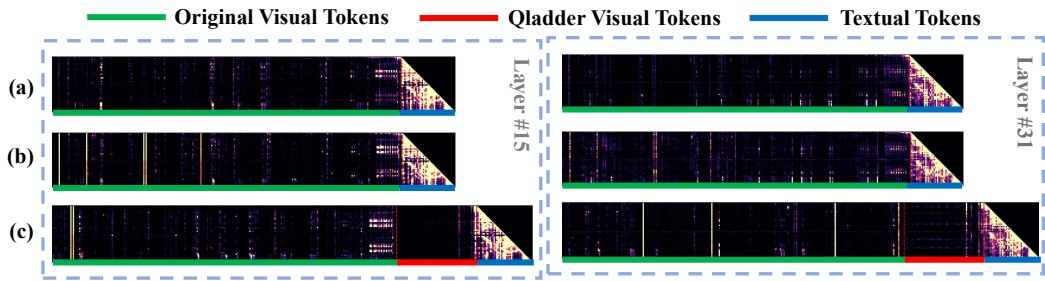

Figure 4: **Visualization of attention maps**. We compare the attention maps in different layer of LLM between different composition, include **(a)** Baseline, **(b)**Baseline+MM-LoRA, and **(c)** Baseline+MM-LoRA+QLadder. Higher brightness indicates higher attention values, with the x-axis representing all tokens, and the y-axis containing only the generated text tokens.

Table 4: Ablation of $\beta$ and $\gamma$ in MM-LoRA. The default rank is set to 256, while $\beta$ and $\gamma$ are used to control the rank values in visual and language LoRA components, respectively.

| Method | RANK | | TextVQA | ScienceQA | MMBench | MME |
|---|---|---|---|---|---|---|
| | $\beta$ | $\gamma$ | | | | |
| LoRA | - | - | 58.1 | 69.1 | 63.8 | 1460 |
| MMLoRA | 1 | 0 | $51.2_{(-6.9)}$ | $65.8_{(-3.3)}$ | $56.4_{(-7.4)}$ | $1356_{(-104)}$ |
| | 0.75 | 0.25 | $\mathbf{58.7}_{(+0.6)}$ | $68.6_{(-0.5)}$ | $63.3_{(-0.5)}$ | $1465_{(+5.0)}$ |
| | 0.5 | 0.5 | $58.5_{(+0.4)}$ | $70.1_{(+1.0)}$ | $64.4_{(+0.6)}$ | $1483_{(+23)}$ |
| | 0.25 | 0.75 | $\mathbf{58.7}_{(+0.6)}$ | $\mathbf{71.2}_{(+2.1)}$ | $64.8_{(+1.0)}$ | $\mathbf{1500}_{(+40)}$ |
| | 0 | 1 | $57.9_{(-0.2)}$ | $70.1_{(+1.0)}$ | $\mathbf{65.4}_{(+1.6)}$ | $1480_{(+20)}$ |

Table 5: Ablation of query number in QLadder. $N_q$ represents the number of learnable query.

| Method | $N_q$ | ScienceQA | MMBench | MME |
|---|---|---|---|---|
| baseline | - | 69.1 | 63.8 | 1460 |
| +QLadder | 16 | $70.4_{(+1.3)}$ | $63.9_{(+0.1)}$ | $1481_{(+21)}$ |
| | 32 | $70.6_{(+1.5)}$ | $64.6_{(+0.8)}$ | $1493_{(+33)}$ |
| | 64 | $\mathbf{71.2}_{(+2.1)}$ | $\mathbf{64.8}_{(+1.0)}$ | $\mathbf{1500}_{(+40)}$ |
| | 128 | $69.7_{(+0.6)}$ | $64.2_{(+0.4)}$ | $1473_{(+13)}$ |

MLLMs. However, introducing $\gamma$ greatly improves performance, demonstrating that learning both vision and language simultaneously accelerates modality alignment. When $\gamma$ is set to 1, there is a slight performance decline, but MM-LoRA still matches LoRA's performance, suggesting that visual token learning is less critical than language token learning in LLMs. This indicates that during the instruction tuning phase of MLLM training, more emphasis should be placed on learning language tokens. Furthermore, when both $\beta$ and $\gamma$ are set to 0.5, the performance of MM-LoRA significantly outperforms LoRA. This intuitively demonstrates that the multimodal decoder can avoid interference between modalities by separating them, thus significantly enhancing the performance of MLLMs.

**QLadder in Vision Encoder.** To validate the effectiveness of QLadder and determine the optimal number of queries, we conducted experiments with QLadder. The results, shown in Table 5, indicate that the inclusion of QLadder significantly enhances our model's performance. This demonstrates that even with a slight increase in visual tokens, without introducing a new visual encoder, the model's visual recognition capabilities can be improved. As the number of queries increased, our model's performance gradually improved, reaching its best performance with 64 queries. However, further increasing the number of queries led to a performance decline, indicating that too many queries can negatively impact the model's performance. To explicitly demonstrate the computational costs and efficiency of MLLMs with and without QLadder, we tested the memory usage and inference speed under both setting. As shown in Table 8, even with QLadder, MLLMs only increase memory usage by 0.582G, and the inference speed decreases by just 0.11 tokens/s. This shows that the additional computational costs and efficiency impacts of QLadder are minimal and acceptable given the improvements it brings.

**QLadder tuning v.s. Visual Encoder tuning.** To explore the impact of fine-tuning QLadder versus directly fine-tuning the Visual Encoder, we conducted comparative experiments to evaluate the effects of tuning the vision encoder, freezing the vision encoder, and adding Q-Ladder. The results are shown in Table 7. Tuning the Vision Encoder often leads to the loss of pre-trained knowledge and does not significantly enhance MLLM's performance. In some benchmark tests, it may even have a negative impact. Freezing the Vision Encoder preserves pre-trained knowledge but lacks further optimization potential. Adding Q-Ladder significantly improves MLLM's performance by enhancing visual feature representation with a small number of additional visual tokens, while retaining pre-trained knowledge.

Table 6: Comparision with QLadder and additional Visual Encoder. To explore the performance in visual grounding ability, we selected MMVP, POPE, MMBench, and TextVQA for experiments. The data used in the experiments is consistent with that of LLaVA-v1.5.

| Method | Size | add visual tokens | MMVP | POPE | MMBench | TextVQA |
|---|---|---|---|---|---|---|
| LLaVA-v1.5 | 7B | - | 24.0 | 85.9 | 64.3 | 58.2 |
| LLaVA-v1.5 + MOF Tong et al. (2024) | 7B | 256 | $27.1_{(+3.1)}$ | $86.2_{(+0.3)}$ | $60.1_{(-4.2)}$ | $56.5_{(-1.7)}$ |
| LLaVA-v1.5 + QLadder | 7B | 64 | $27.6_{(+3.6)}$ | $86.5_{(+0.6)}$ | $66.3_{(+2.0)}$ | $58.8_{(+0.6)}$ |
| LLaVA-v1.5 | 13B | - | 24.7 | 85.9 | 67.7 | 61.3 |
| LLaVA-v1.5 + MOF Tong et al. (2024) | 13B | 256 | $28.0_{(+3.3)}$ | $86.3_{(+0.4)}$ | $61.6_{(-6.1)}$ | $55.3_{(-6.0)}$ |
| LLaVA-v1.5 + MOF Tong et al. (2024) | 13B | 576 | $31.2_{(+6.5)}$ | $86.7_{(+0.8)}$ | $65.4_{(-2.3)}$ | $58.7_{(-2.6)}$ |

These results demonstrate that Q-Ladder effectively strengthens visual feature representation and avoids the negative effects associated with tuning the vision encoder.

Table 7: Comparing different tuning strategies for visual encoders.

| Method | Vision Encoder | TextVQA | MMBench | MM-Vet |
|---|---|---|---|---|
| baseline | freezing | 58.1 | 64.1 | 31.5 |
| baseline | tuning | 57.7 | 64.3 | 31.1 |
| baseline | add QLadder | **58.8** | **66.3** | **33.7** |

Table 8: Comparison of computational load and resource utilization during inference.

| Setting | Memory used | Inference Speed (token/s) |
|---|---|---|
| Arcana (w/o QLadder) | 15.243GB | 22.58 |
| Arcana (w QLadder) | 15.825GB | 22.47 |

**QLadder v.s. additional Visual Encoder.** Recently, there has been works exploring the addition of extra visual encoders to achieve better visual representations, *e.g.*, MOF Tong et al. (2024), which uses Dinov2 Oquab et al. (2024) as a second visual encoder to enhance the grounding ability of MLLMs. To explore the impact of adding QLadder and adding extra visual encoder, we conducted detailed experiments to directly compare Q-Ladder with the MoF method, which integrates DINOv2, under the LLaVA-v1.5 setting. The results are shown in Table 6. Our experimental results show that both Q-Ladder and MoF performed well in visual grounding, achieving significant improvements on the MMVP and POPE benchmarks. However, MoF's performance declined on more comprehensive benchmarks like MMbench and OCR benchmarks like TextVQA. This decline is primarily due to MoF's reliance on DINOv2 for visual grounding, which, while enhancing grounding capabilities, weakened visual understanding, leading to poorer results on MMbench and TextVQA. Additionally, the integration of DINOv2 significantly increased the model's training time. In contrast, Q-Ladder enhances both visual grounding and visual understanding through adaptive learning of distinguishing features. This dual improvement allows Q-Ladder to maintain or boost performance across a wide range of benchmarks, even when using a smaller dataset (over 2 million samples from Arcana). This is why Q-Ladder continues to achieve performance gains across various benchmarks, including comprehensive and OCR benchmarks.

**Impact of MM-LoRA and QLadder in MLLMs.** To investigate the impact of MM-LoRA and QLadder in multimodal scenarios, we visualized the attention maps of Arcana with and without these modules in MM-Vet benchmark Yu et al. (2023). The visualization results, shown in Fig. 4, display the attention scores of generated tokens over the input sequence during the generation process. It can be seen that MLLM decoder initially focuses more on text tokens and gradually increases attention to visual tokens in the middle and subsequent layers. This indicates that visual and language information play different roles in MLLMs. The discussion about shallow-level attention maps, which also reflects this point, is provided in the Appendix. Additionally, with MM-LoRA, we observe a significant increase in attention to visual tokens in the middle and subsequent layers, indicating that MM-LoRA helps prevent information confusion and promotes cooperation between different modalities. With the introduction of QLadder, the MLLM decoder shows increased attention to visual tokens across all layers. The highlighted regions of visual tokens further indicate that QLadder not only enhances the model's focus on visual tokens but also enriches the visual information, achieving optimal performance in multimodal tasks.

**Visualization results.** To showcase Arcana's outstanding performance in visual perception, we visualized its performance across various types of multimodal tasks. As illustrated in Fig. 5, visual perception information is highlighted in orange. In detailed description tasks, our model not only accurately identifies and describes low-level visual information such as colors and textures for each

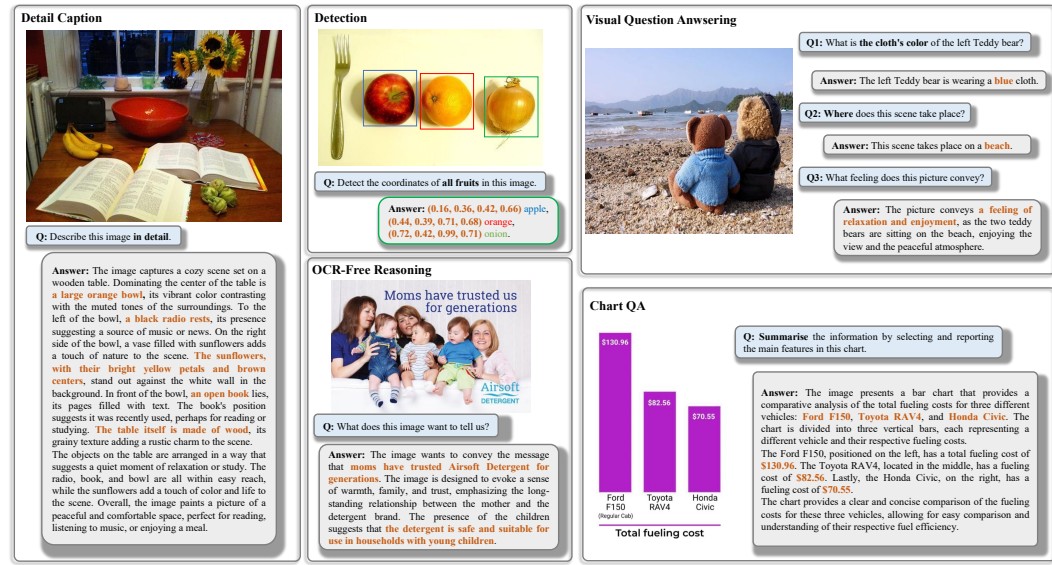

Figure 5: Examples of results generated by Arcana were sampled, focusing on tasks that test visual perception capabilities, such as detailed captions, detection, and OCR-reasoning. In the answers, all visual recognition-related responses are highlighted in orange.

object in the image but also precisely recognizes and describes high-level visual information such as positions and relationships of each object. Moreover, detection tasks further demonstrate our model's effectiveness in visual recognition and localization. OCR-Free inference and chart-based question answering tasks not only exhibit our model's OCR recognition capabilities but also demonstrate its reasoning prowess. Visual question answering tasks showcase our model's excellent multi-turn dialogue capabilities on the foundation of precise identification.

In summary, Arcana utilizes a multimodal decoder to avoid information interference between different modalities. QLadder offers an innovative strategy for enhancing visual representations with limited data. By adding a small number of visual tokens, it significantly improves the performance of large multimodal language models. This finding is significant for the future of multimodal model, as it presents an effective approach to achieving notable performance improvements even with limited data resources. By combining these techniques, future multimodal models will handle complex tasks with greater flexibility and efficiency.

## 5 CONCLUSION

In this paper, we introduce a new multimodal large language model, **Arcana**, which incorporates two novel techniques. Unlike current mainstream methods, Arcana employs MM-LoRA for a multimodal decoder, enabling more efficient information processing and integration across different modalities. MM-LoRA effectively combines data from various modalities without significantly increasing computational complexity, reducing information interference between modalities. Secondly, we present the QLadder structure, which demonstrates for the first time that with limited multimodal training data, retaining the capabilities of a pre-trained model and adding a small number of visual encoders can still enhance the performance of multimodal language models. This hierarchical structure progressively refines and enhances the expression of visual information, resulting in improved adaptability and generalization in multimodal tasks. With these two key techniques, Arcana not only excels in multimodal tasks but also shows potential for performance improvement even in data-constrained environments. Additionally, the severe lack of visual information in the image captions of open-source data limits the visual perception capabilities of multimodal large language models. To address this, we designed a data engine that uses diverse visual annotation models and large language models to generate captions rich in visual information.

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
