# Improving Multi-modal Large Language Model through Boosting Vision Capabilities – Supplementary Material

## A  Appendix

### A.1  Detailed Evaluation Results.

**POPE.** We conduct the hallucination evaluation using POPE Li et al. (2023), the results are shown in Table. From the results in the Table 1, we can find Arcana achieves higher F1 scores on the popular and adversarial split, showing the robustness of our model in terms of object hallucination compared to other MLLMs.

**MMBench.** MMBench Liu et al. (2023b) is used to evaluate the model's ability of Perception and Reasoning. The detail results for various MLLMs are presented in Table 2.

### A.2  More visualization results.

To demonstrate the effectiveness and generalization ability of Arcana, we provide more qualitative results in Fig. 1. We visualize its performance across various types of multimodal tasks, including Detail Caption, Detection, Knowledge, OCR-Free Reasoning, Visual Question Answering and ChartQA. To investigate the impact of MM-LoRA and QLadder in multimodal scenarios, we visualized the attention maps of different layers with and without these modules in Fig. 2.

### A.3  Broader Impact

This paper present Arcana, which target at improving the visual understanding capability for boosting the vision-language models. To achieve this goal, Arcana conducts a series of explorations into visual learning within the model structure. On one hand, Arcana demonstrates that decoupling the learning of visual and language representation within the LLM is beneficial for avoiding information confusion while preserving the uniqueness of each modality, and based on this, proposes MM-LoRA. On the other hand, Arcana asserts that under limited training data, it is important to retain the pre-trained image encoder's capabilities and introduces QLadder, which incorporates a small number of visual tokens to enhance the model's learning and representation abilities for visual information. Extensive experiments demonstrate the effectiveness and generalization ability of Arcana.

The positive societal impacts of the work include:

- **Improved Human-Machine Interaction**: Enhanced visual perception in multimodal models can lead to more intuitive and effective human-machine interactions. This could improve applications such as virtual assistants, customer service bots, and educational tools, making them more responsive and capable of understanding complex visual contexts.

- **Advancements in AI Research**: The Arcana model's innovative architecture and data handling approaches could stimulate further research in the AI community, leading to new breakthroughs and applications in various fields, from healthcare to autonomous vehicles, where precise visual perception is crucial.

- **Better Performance in Real-World Applications**: By addressing the deficiencies in low-level and high-level visual perception, Arcana can improve performance in practical applications like object detection in surveillance, quality control in manufacturing, and detailed image analysis in medical diagnostics.

The negative societal impacts may include:

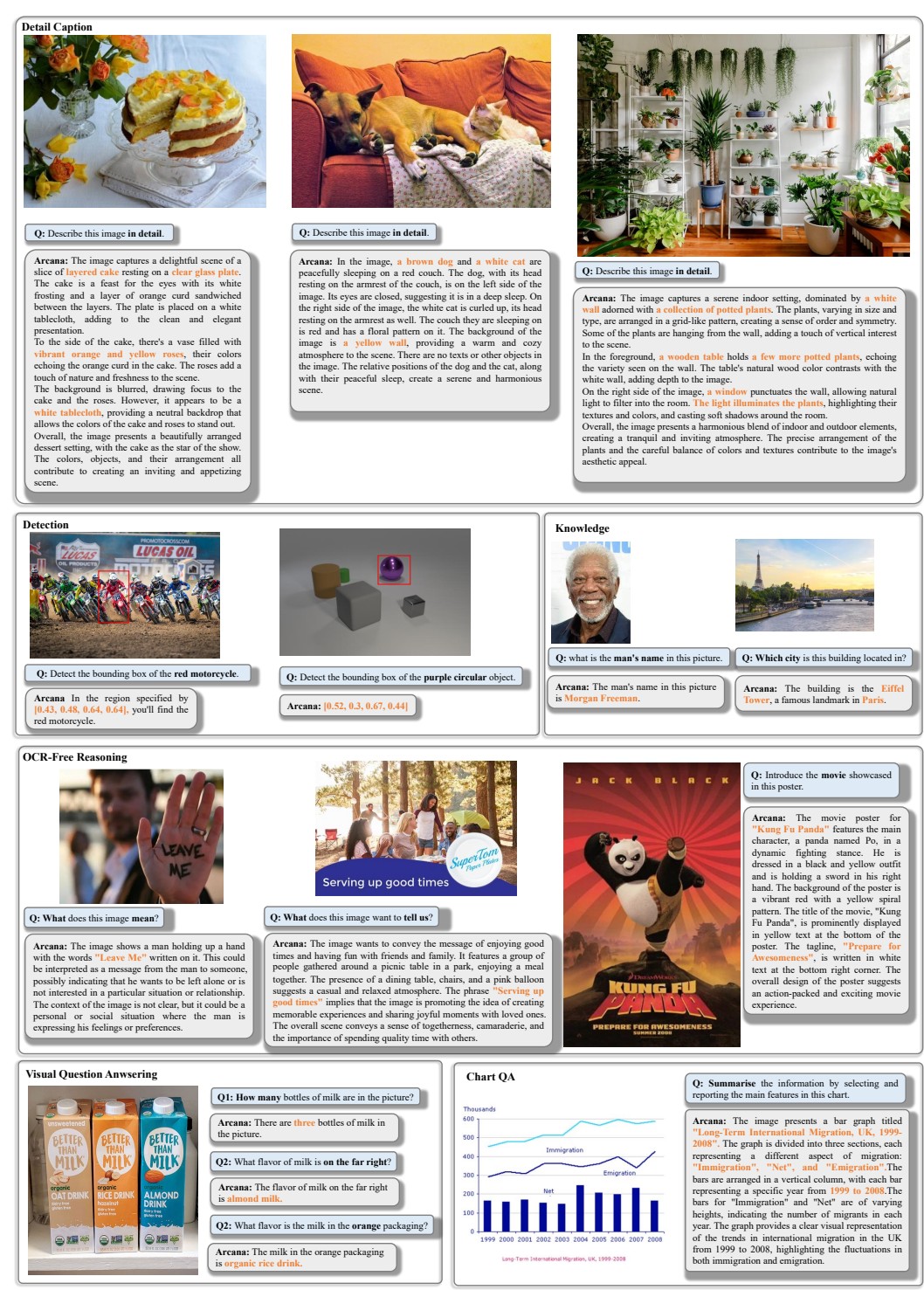

Figure 1: More qualitative results. Main feature in answer is highlight in orange.

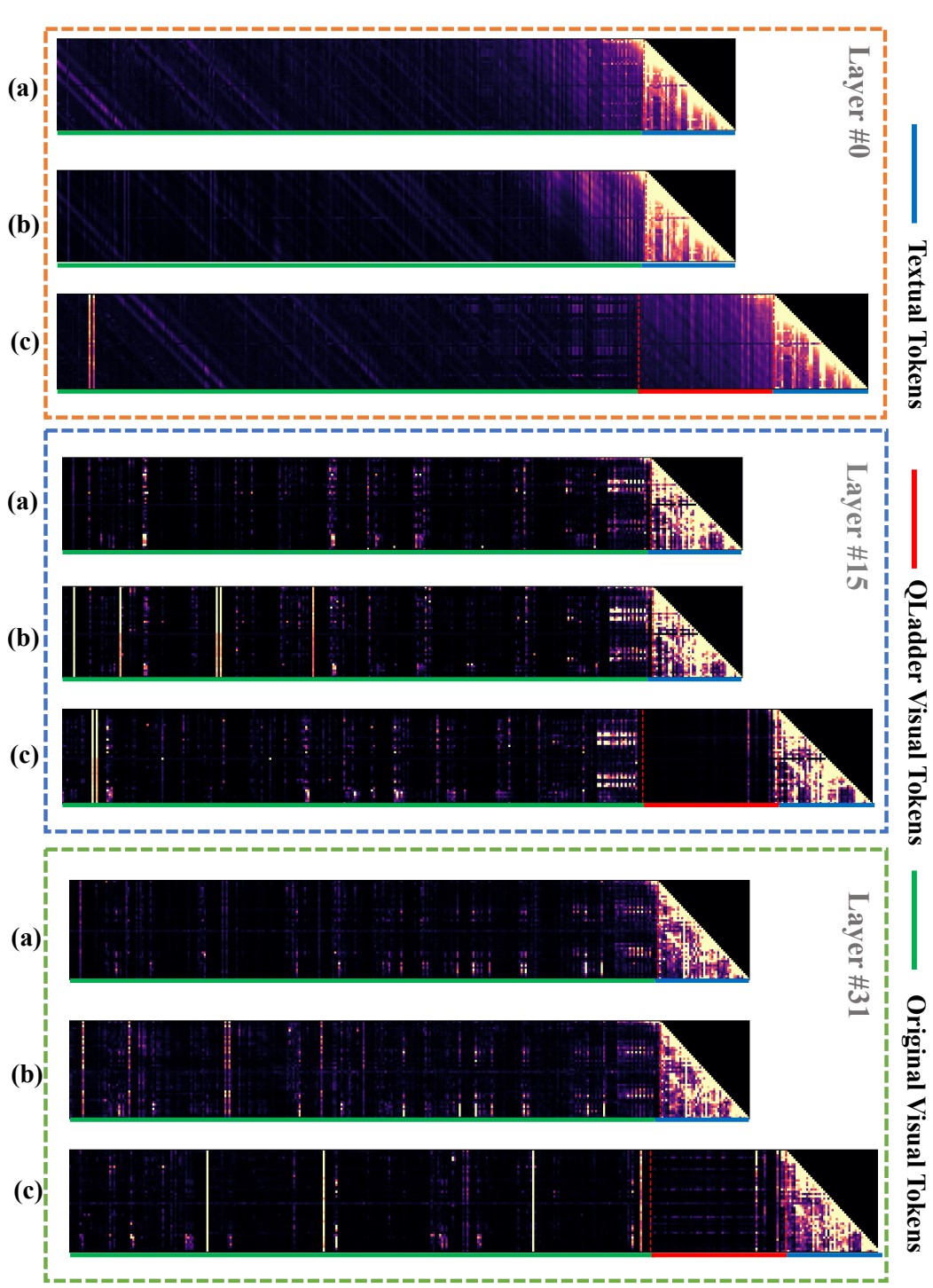

Figure 2: **Visualization of attention maps**. We compare the attention maps in different layer of LLM between different composition, include **(a)** Baseline, **(b)**Baseline+MM-LoRA, and **(c)** Baseline+MM LoRA+QLadder. Higher brightness indicates higher attention values, with the x-axis representing all tokens, and the y-axis containing only text tokens.

Table 1: Object hallucination benchmark using POPE evaluation pipeline. "Yes" signifies the likelihood of the model producing a positive response.

| Datasets | Metrics | Arcana (Ours) | mPLUG-Owl2 Ye et al. (2023) | LLaVA-v1.5 Liu et al. (2023a) | Shikra Chen et al. (2023) | InstructBLIP Dai et al. (2024) | MiniGPT-4 Zhu et al. (2023) |
|---|---|---|---|---|---|---|---|
| Random | Accuracy (↑) | 88.87 | 88.28 | 88.38 | 86.90 | 88.57 | 79.67 |
| | Precision (↑) | 96.59 | 94.34 | 96.56 | 94.40 | 84.09 | 78.24 |
| | Recall (↑) | 81.27 | 82.20 | 80.33 | 79.27 | 95.13 | 82.20 |
| | F1-Score (↑) | 88.27 | 87.85 | 87.70 | 86.19 | 89.27 | 80.17 |
| | Yes (→ 50%) | 43.37 | 44.91 | 42.89 | 43.26 | 56.57 | 52.53 |
| Popular | Accuracy (↑) | 88.07 | 86.20 | 87.67 | 83.97 | 82.77 | 69.73 |
| | Precision (↑) | 94.06 | 89.46 | 94.14 | 87.55 | 76.27 | 65.86 |
| | Recall (↑) | 81.27 | 82.06 | 80.33 | 79.20 | 95.13 | 81.93 |
| | F1-Score (↑) | 87.20 | 85.60 | 86.69 | 83.16 | 84.66 | 73.02 |
| | Yes (→ 50%) | 43.20 | 45.86 | 42.67 | 45.23 | 62.37 | 62.20 |
| Adversarial | Accuracy (↑) | 86.57 | 84.12 | 85.23 | 83.10 | 72.10 | 65.17 |
| | Precision (↑) | 90.90 | 85.54 | 89.06 | 85.60 | 65.13 | 61.19 |
| | Recall (↑) | 81.27 | 82.13 | 80.33 | 79.60 | 95.13 | 82.93 |
| | F1-Score (↑) | 85.81 | 83.80 | 84.47 | 82.49 | 77.32 | 70.42 |
| | Yes (→ 50%) | 44.70 | 48.00 | 45.10 | 46.50 | 73.03 | 67.77 |

- **Privacy Concerns**: Enhanced visual perception capabilities may lead to more invasive surveillance technologies. The ability to detect and interpret small objects and detailed visual information could be misused to infringe on individuals' privacy, leading to unauthorized tracking and monitoring.

- **Security Risks**: Advanced visual perception models could be exploited for malicious purposes, such as by enhancing the capabilities of autonomous weapons or by improving the precision of surveillance systems used by authoritarian regimes to suppress dissent.

- **Dependence on Technology**: Increasing reliance on advanced AI for visual tasks may lead to a decrease in human skills and awareness in certain fields. Over-dependence on such technology without proper human oversight could have negative implications for critical decision-making processes.

Table 2: CircularEval multi-choice accuracy results on MMBench Liu et al. (2023b) dev set. We adopt the following abbreviations: LR for Logical Reasoning; AR for Attribute Reasoning; RR for Relation Reasoning; FP-C for Fine-grained Perception (Cross Instance); FP-S for Finegrained Perception (Single Instance); CP for Coarse Perception.

| Method | Language Model | Vision Model | Overall | LR | AR | RR | FP-S | FP-C | CP |
|---|---|---|---|---|---|---|---|---|---|
| MiniGPT-4 Zhu et al. (2023) | Vicuna-7B | EVA-G | 12.0 | 13.6 | 32.9 | 8.9 | 28.8 | 11.2 | 28.3 |
| InstructBLIP Dai et al. (2024) | Vicuna-7B | EVA-G | 33.9 | 21.6 | 47.4 | 22.5 | 33.0 | 24.4 | 41.1 |
| LLaMA-Adapter-v2 Gao et al. (2023) | LLaMa-7B | CLIP ViT-L/14 | 38.9 | 7.4 | 45.3 | 19.2 | 45.0 | 32.0 | 54.0 |
| LLaVA Liu et al. (2024) | Vicuna-7B | CLIP ViT L/14 | 36.2 | 15.9 | 53.6 | 28.6 | 41.8 | 20.0 | 40.4 |
| Shikra Chen et al. (2023) | Vicuna-7B | CLIP ViT-L/14 | 60.2 | 33.5 | 69.6 | 53.1 | 61.8 | 50.4 | 71.7 |
| LLaVA-v1.5 Liu et al. (2023a) | Vicuna-7B | CLIP ViT-L/14 | 64.3 | 33.1 | 69.3 | 57.4 | 68.9 | 54.5 | 76.4 |
| mPLUG-Owl2 Ye et al. (2023) | LLaMA2-7B | CLIP ViT-L/14 | 65.4 | 29.2 | **69.7** | 61.7 | 67.0 | **60.0** | 79.5 |
| **Arcana (Ours)** | Vicuna-7B | CLIP ViT-L/14 | **67.4** | **34.7** | 69.3 | **62.6** | 69.6 | 58.7 | 83.1 |

In summary, while the Arcana model holds promise for significant advancements and positive contributions to society, it is crucial to address the associated risks through responsible development, deployment, and regulation to mitigate potential negative impacts.

## A.4 LIMITATIONS AND FUTURE WORK

The previous experiments have demonstrated the effectiveness of Arcana. Although the multimodal decoder has proven effective, giving each modality its own learning space significantly increases the number of parameters. While MM-LoRA only adds a small number of parameters to achieve a multimodal decoder, the independent LoRA parameters for different modalities cannot be merged into the LLMs' weights, thereby increasing inference costs. The introduction of QLadder enhances visual representation capabilities, but it comes at the cost of adding visual tokens, which also increases inference costs. Additionally, compared to existing state-of-the-art methods, we used only about 2M training data, limiting Arcana's performance.

To further unlock Arcana's potential, we will design a more efficient multimodal decoder that improves performance while reducing inference costs. We will also focus on designing a more efficient visual encoder that uses fewer visual tokens to represent visual features, enhancing training efficiency and reducing inference costs. Finally, we plan to leverage our data engine to annotate more high-quality caption data to fully unleash Arcana's potential.