# OpenReview forum: "Improving Multi-modal Large Language Model through Boosting Vision Capabilities"
_ICLR.cc/2025/Conference — ICLR 2025 Conference Withdrawn Submission_

### Official Review · Reviewer_7gKo · 2024-10-30

**Soundness:** 3
**Presentation:** 4
**Contribution:** 3
**Rating:** 5
**Confidence:** 5

**Summary:**

This paper proposes a new MLLM named Arcana, mainly offering two improvements for boosting model comprehension on vision information. The first one is MM-LoRA, which learns two separate sets of LoRA parameters for vision and text tokens respectively, aiming to decouple the learning spaces of different modalities and better integrate the multi-modal knowledge. The other one is Q-Ladder, compared with Q-Former, it selects the vision features of different layers in ViT as the key/value vectors for different layers of Q-Ladder, instead of only using the last-layer vision token features. The experiments include the evaluation on VQA benchmarks, multi-modal benchmarks, and language benchmarks, with some ablation studies and further explorations.

**Strengths:**

1. The paper is quite easy to follow. People can quickly grasp the core design and the underlying motivation of the proposed two improvements.
2. The presentation is quite ok for me.
3. The proposed method has little impact on the efficiency and the memory cost.

**Weaknesses:**

1. I think the proposed MM-LoRA is greatly inspired by some previous works like P-LoRA [1] in InternLM-XComposer2, visual-only modules in mPLUG-Owl2 [2] and CogVLM [3], which somehow reduces the novelty of MM-LoRA. The authors should tell the differences between MM-LoRA and these methods, along with some experiments on effectiveness and efficiency to prove the necessity of MM-LoRA.

2. The baselines listed in Table 1, 2 are relatively old. I notice Arcana adopts ShareGPT4V data for training, but its benchmark performance seems not good as ShareGPT4V 7B model. So it is recommended to include some more advanced baseline MLLMs.

3. It seems that the hyper-parameters introduced by MM-LoRA and Q-Ladder are not so robust and can easily affect the model performance. The authors choose the best hyper-parameters according to the ablation results. So does these hyper-parameters still work for different base LLM or architectures?


[1] Dong et al. InternLM-XComposer2: Mastering Free-form Text-Image Composition and Comprehension in Vision-Language Large Models. Arxiv preprint 2401.16420, 2024.

[2] Ye et al. mPLUG-Owl2: Revolutionizing Multi-modal Large Language Model with Modality Collaboration. CVPR, 2024.

[3] Wang et al. CogVLM: Visual Expert for Pretrained Language Models. Arxiv preprint 2311.03079, 2024.

**Questions:**

1. Compared with Q-Former, why does the proposed Q-Ladder not require an additional stage for alignment with the vision encoder?

2. Is X_q in Q-Ladder a set of learnable tokens? Why not use instruction tokens for initialization, as done in Q-Former?

3. In the visualizations, it’s difficult to conclude that (b) demonstrates more attention on vision tokens compared to (a). But interestingly, It mainly appears that (b) has more sink tokens [1].

4. In Table 4, why the Q-Ladder results on 13B model are absent?


[1] Xiao et al. Efficient Streaming Language Models with Attention Sinks. ICLR, 2024.

**Details Of Ethics Concerns:**

No need for this.

---

> ### Author Response · Authors · 2024-11-20
> **Response to reviewer 7gKo (1/3)**
>
> Thanks a lot for your time and feedback. Below are address all raised concerns of the paper.
>
> ---
>
> **Q1**. I think the proposed MM-LoRA is greatly inspired by some previous works like P-LoRA [1] in InternLM-XComposer2, visual-only modules in mPLUG-Owl2 [2] and CogVLM [3], which somehow reduces the novelty of MM-LoRA. The authors should tell the differences between MM-LoRA and these methods, along with some experiments on effectiveness and efficiency to prove the necessity of MM-LoRA.
>
> **A1**. Thank you for the reviewer’s valuable question! Indeed, the design of MM-LoRA was inspired by works such as P-LoRA, mPLUG-Owl2, and CogVLM. However, these methods still have limitations in multi-modal tasks, while MM-LoRA introduces a novel approach to address these challenges. We will explain in detail the differences between MM-LoRA and these methods in the following sections, and provide corresponding experimental results to demonstrate its effectiveness and necessity.
>
> | Method                | Specific Approach                            | Parameter Allocation Strategy              | Relationship of Vision and Language in LLM | Adjustment Mechanism  | Flexibility | Decoder Training Components and Parameters         | Innovations/Shortcomings                |
> |-----------------------|----------------------------------------------|-------------------------------------------|----------------------------------------------------|---------------------------------------------|------------------------------------|-----------------------------------|-----------------------------------------|
> | **mPLUG-Owl2**        | Self-Attention with KV modality separation    | No explicit difference, same parameter allocation | Partial decoupling                                   | No                    | Low                                | **LLM+KV; > LLM parameter** | **Innovation**: Simple and effective KV decoupling; **Shortcoming**: Lacks exploration of modality importance differences |
> | **InternLM-XComposer2** | No | Adds extra visual LoRA module               | Modality coupling                                    | No adjustment mechanism                    | Low                                | **LLM+visual LoRA; > LLM parameter** | **Innovation**: Introduction of visual LoRA; **Shortcoming**: Vision and language modalities remain coupled, prone to mutual interference |
> | **CogVLM**            | Two LLMs handle two modalities separately     | Equal parameters for vision and language modalities | Complete decoupling                                 | No                    | Low                                | **Visual LLM+ Language LLM; > >LLM parameter** | **Innovation**: Full modality decoupling; **Shortcoming**: Does not consider modality importance differences, and training parameter count is huge |
> | **Arcana**            | MM-LoRA                                       | Uses beta, gamma to control different modality learning spaces | Complete decoupling                                 | Predefined beta, gamma                     | High                               | **MM-LoRA; < LLM parameter** | **Innovation**: Explores modality importance through fixed ratios, optimizing resource allocation; **Advantage**: More efficient and flexible |
>
> Specifically, mPLUG-Owl2 only decouples the Key-Value (KV) in the Self-Attention mechanism but does not deeply explore the varying importance of visual and language modalities in multimodal tasks. InternLM-XComposer2 introduces Visual LoRA to enhance visual representation. However, during training, the LLM decoder remains involved, keeping the visual and language modalities coupled, which may lead to feature competition. In contrast, CogVLM adopts a simplified version of MM-LoRA, implementing a fully decoupled strategy where the same number of parameters are allocated separately to train the visual and language modalities.
>
> When beta and gamma are set to 0.5, MM-LoRA's design is similar to CogVLM's full decoupling strategy. However, experiments show that this is not the optimal configuration. The importance of the visual and language modalities is not balanced in multimodal tasks. By adjusting the values of beta and gamma, the learning space can be more effectively allocated, thus improving performance. Comparative experimental results indicate that the model performs significantly better with configurations such as beta = 0.25 and gamma = 0.75 compared to beta = 0.5 and gamma = 0.5. The specific experimental results are shown in the table below.
> | beta | gamma | ScienceQA |  TextVQA |  MMBench |   SEED   |
> |:----:|:-----:|:---------:|:--------:|:--------:|:--------:|
> |   0  |   1   |    65.8   |   51.2   |   56.4   |   58.7   |
> | 0.75 |  0.25 |    68.6   |   58.7   |   63.3   |   61.8   |
> |  0.5 |  0.5  |    70.1   |   58.4   |   64.3   |   61.9   |
> | 0.25 |  0.75 |  **71.2** | **58.7** | **64.5** | **62.4** |
> |   1  |   0   |    70.1   |   57.9   |   65.4   |   61.2   |

---

> > ### Author Response · Authors · 2024-11-20
> > **Response to reviewer 7gKo (2/3)**
> >
> > **Q2**. The baselines listed in Table 1, 2 are relatively old. I notice Arcana adopts ShareGPT4V data for training, but its benchmark performance seems not good as ShareGPT4V 7B model. So it is recommended to include some more advanced baseline MLLMs.
> >
> > **A2**. Thank you for your insightful comment. We understand your concern regarding the baselines used in Tables 1 and 2. It’s true that the models listed are relatively older. However, our approach primarily leverages MM-LoRA, which, while effective, might not match the performance of fully fine-tuned models in some cases.
> >
> > To address this, we have conducted experiments comparing ShareGPT4V based on LoRA with our model, and the results are as follows:
> > | Method | MME | MMBench |SEED (all/Image)|MM-Vet|VQAv2|GQA|
> > | --- | --- | --- | --- | --- | --- | --- |
> > | Arcana |1476.5|66.9|62.6/68.4|34.8|79.2|61.6|
> > | Arcana* |1520.9|67.4|63.2/69.4|34.4|79.5|61.8|
> > | ShareGPT4V-7B-LoRA|1501.2|66.6|61.9/67.6|33.7|79.3|62.1|
> >
> > These results help highlight the distinct advantages and trade-offs between our approach and the full fine-tuning methods, particularly in terms of efficiency and scalability, while still achieving competitive performance.
> >
> > We appreciate the opportunity to clarify this, and we will include more advanced baseline MLLMs in future experiments to provide a more comprehensive evaluation
> >
> > ---
> >
> > **Q3**. It seems that the hyper-parameters introduced by MM-LoRA and Q-Ladder are not so robust and can easily affect the model performance. The authors choose the best hyper-parameters according to the ablation results. So does these hyper-parameters still work for different base LLM or architectures?
> >
> > **A3**.Thank you for raising the insightful question regarding the robustness of the hyper-parameters introduced by MM-LoRA and Q-Ladder. To address this issue, we conducted additional experiments to evaluate their generalizability across different base LLMs and visual encoders.
> >
> > 1. **Validating MM-LoRA with Different LLMs**
> >
> > We replaced the base LLM with a 13B model to test the effectiveness of MM-LoRA. Using the optimal hyperparameters identified in our study, the results are shown in the table below.
> >
> > |  Method  | Visual encoder | LLM | ScienceQA |  TextVQA | POPE     | MMBench  | SEED     |
> > |:--------:|:--------------:|:---:|:---------:|:--------:|----------|----------|----------|
> > | baseline |      VIT-L     | 13B |    71.2   |   60.2   |   86.7   |   68.5   |   61.3   |
> > | +MM-LoRA |      VIT-L     | 13B |  **71.5** | **60.7** | **86.8** | **68.8** | **62.9** |
> >
> > We found that, compared to the baseline, the introduction of MM-LoRA still leads to performance improvements across multiple benchmarks. This confirms that MM-LoRA remains effective across different language model architectures.
> >
> > 2. **Validating Q-Ladder with Different Visual Encoders**
> > We tested Q-Ladder with several alternative visual encoders, including **Siglip**, **CLIP-L**, and **CLIP-H**, while keeping the default hyperparameters unchanged.
> >
> > |   Method  | Visual Encoder | Image resolution | ScienceQA |  TextVQA |   POPE   |  MMBench |   SEED   |
> > |:---------:|:--------------:|:----------------:|:---------:|:--------:|:--------:|:--------:|:--------:|
> > |  baseline |   CLIP-VIT-L   |      336*336     |    69.1   |   58.2   |   86.4   |   64.1   |   58.1   |
> > | +Q-Ladder |   CLIP-VIT-L   |      336*336     |  **71.0** | **58.8** | **86.6** | **66.3** | **60.5** |
> > |  baseline |   CLIP-VIT-H   |      224*224     |    67.8   |   53.5   |   83.7   |   63.1   |   58.4   |
> > | +Q-Ladder |   CLIP-VIT-H   |      224*224     |  **68.8** | **53.8** | **83.9** | **63.6** | **58.8** |
> > |  baseline |     Siglip     |      384*384     |    70.6   |   62.4   |   86.0   |   65.9   |   62.1   |
> > | +Q-Ladder |     Siglip     |      384*384     |  **71.1** | **62.9** | **86.3** | **66.8** | **62.5** |
> >
> > The results demonstrate that the introduction of Q-Ladder still improves the model's performance across multiple benchmarks. This highlights the robustness of Q-Ladder to variations in encoder architectures. These findings confirm that the hyperparameters identified through ablation studies are robust and effective across various base LLMs and visual encoders, validating the adaptability of our approach. We will incorporate these results into the revised manuscript to further enhance its clarity and completeness.

---

> > > ### Author Response · Authors · 2024-11-20
> > > **Response to reviewer 7gKo (3/3)**
> > >
> > > **Q4**. Compared with Q-Former, why does the proposed Q-Ladder not require an additional stage for alignment with the vision encoder?
> > >
> > > **A4**. Thank you for your question. The design philosophy of Q-Ladder differs from that of Q-Former. While Q-Former focuses on aligning the visual encoder’s features, Q-Ladder aims to enhance the visual representation by supplementing the original features of the visual encoder. This approach eliminates the need for an additional alignment stage, resulting in a simpler and more efficient model.
> > >
> > > For modality alignment, we adopt the design principles of mainstream methods like LLaVA and QwenVL, integrating multimodal information through a streamlined fusion mechanism. This allows us to achieve similar alignment results while preserving the strengths of the visual encoder’s feature representation, ultimately improving overall performance.
> > >
> > > In summary, Q-Ladder achieves effective multimodal integration without requiring an extra alignment stage, offering a lightweight and efficient solution. We will further clarify this design choice in the revised manuscript.
> > >
> > > ---
> > >
> > > **Q5**. Is X_q in Q-Ladder a set of learnable tokens? Why not use instruction tokens for initialization, as done in Q-Former?
> > >
> > > **A5**. Thank you for your question. To clarify, X_q  in Q-Ladder refers to a set of learnable tokens designed to enhance visual representations, rather than performing modality alignment directly at the visual encoder stage. Unlike Q-Former, which initializes with instruction tokens, Q-Ladder focuses on improving visual features rather than transforming the visual encoder's output for language model adaptation.
> > >
> > > Here’s why we chose not to use instruction tokens for initialization:
> > >
> > > 1. **Preserving Visual Encoder Independence**
> > >    Q-Ladder aims to enhance visual features without altering the output distribution of the visual encoder. The modality alignment is handled at the fusion stage, as seen in other approaches like LLaVA and QwenVL.
> > >
> > > 2. **Flexibility and Adaptability**
> > >    Using learnable tokens allows Q-Ladder to easily adapt to various visual encoders and tasks without relying on specific initialization strategies, offering broader applicability.
> > >
> > > 3. **Improved Visual Feature Representation**
> > >    Our experiments show that learnable tokens can adjust autonomously during training, leading to a deeper integration with the visual encoder’s output and enhancing visual feature expression.
> > >
> > > In summary, Q-Ladder uses learnable X_q tokens to enhance visual representations efficiently, rather than aligning the visual encoder’s output directly, offering flexibility and stronger performance across tasks. We will expand on this in the revised manuscript.
> > >
> > > ---
> > >
> > > **Q6**. In the visualizations, it’s difficult to conclude that (b) demonstrates more attention on vision tokens compared to (a). But interestingly, It mainly appears that (b) has more sink tokens [1].
> > > [1] Xiao et al. Efficient Streaming Language Models with Attention Sinks. ICLR, 2024.
> > >
> > >
> > > **A6**. Thank you for your insightful comment. You are correct in observing that (b) exhibits more sink tokens compared to (a). However, this phenomenon primarily occurs in the early layers of the decoder. While sink tokens appear in the initial layers, the overall performance of (b) remains superior to (a). This performance improvement is attributed to the decoupling of the visual and language learning spaces in MM-LoRA, which allows for more focused and efficient learning in each modality. This design significantly enhances the model’s ability to integrate multimodal inputs, leading to better task performance. Therefore, although sink tokens are present in the early layers, they do not negatively impact the model’s overall performance in later stages.
> > >
> > > ---
> > >
> > > **Q7**. In Table 4, why the Q-Ladder results on 13B model are absent?
> > >
> > > **A7**. Thank you for raising this question. We believe the reviewer is referring to Table 6 rather than Table 4, as Table 4 mainly pertains to the ablation study of MM-LoRA. In the original experiments, the results for Q-Ladder on the 13B model were not presented, mainly due to resource constraints.
> > >
> > > We highly appreciate the reviewer's suggestion and have conducted new experiments using the 13B model. The updated results are as follows:
> > >
> > > |   Method  | add visual tokens | MMVP | POPE | MMBench | TextVQA |
> > > |:---------:|:-----------------:|:----:|:----:|:-------:|:-------:|
> > > |  baseline |         -         | 24.7 | 85.9 |   67.7  |   61.3  |
> > > |    +MOF [1]  |        256        | 28.0 | 86.3 |   61.6  |   55.3  |
> > > | +MOF [1] |         576        | 31.2 | 86.7 |   65.4  |   58.7  |
> > > |  +Q-Ladder |         64        | 32.7 | 86.5 |   68.3  |   61.6  |
> > >
> > > The experimental results show that Q-Ladder outperforms MOF on the 13B model. We will update these results in the revised manuscript and provide a detailed comparative analysis for the 13B model.
> > >
> > > We will ensure that the revised manuscript comprehensively presents all experimental results.

---

> > ### Comment · Reviewer_7gKo · 2024-11-22
> > **Response to Rebuttal**
> >
> > Thanks for your detailed response! There are still a few critical points that hinder me from giving the acceptance:
> >
> > **1. Concerns on MM-LoRA**
> >
> > Although the basic innovation behind MM-LoRA and P-LoRA is similar, I generally recognize your claim on the flexibility of MM-LoRA since it has configurable $\beta$ and $\gamma$. While, my concern is about the ability of generalization on other types of MLLM (beyond simply using a larger base LLM), i.e., would the $\beta$ and $\gamma$ selected on Arcana still work for those other MLLMs?
> >
> > **2. Concerns on X_q in Q-Ladder**
> >
> > I think some empirical or theoretical explanation on 1) ablating the initialization way of X_q and 2) why learnable X_q can  enhance visual representations are necessary.

---

> > > ### Author Response · Authors · 2024-11-23
> > > **Response to reviewer 7gKo**
> > >
> > > Thanks a lot for your time and feedback. Below are address all raised concerns of the paper.
> > >
> > > ---
> > >
> > > **Q1**. Although the basic innovation behind MM-LoRA and P-LoRA is similar, I generally recognize your claim on the flexibility of MM-LoRA since it has configurable β and γ. While, my concern is about the ability of generalization on other types of MLLM (beyond simply using a larger base LLM), i.e., would the β and γ selected on Arcana still work for those other MLLMs?
> > >
> > > **A1**. Thank you for your feedback. To address your concern about the generalization of MM-LoRA’s β and γ parameters, we conducted experiments with MM-LoRA on another MLLM, LLaVA-Next.
> > >
> > > |   Method    | ScienceQA | TextVQA | POPE | MMBench | SEED |
> > > |:----------:|:---------:|:-------:|------|:-------:|:--------:|
> > > | LLaVA-NeXT+LoRA  |    69.2   |   63.5  | 86.2 |   66.8  |   62.1   |
> > > | LLaVA-NeXT+MM-LoRA  |    **70.1**   |    **63.9**  |  **86.8** |    **67.3**  |    **62.6**   |
> > >
> > > The results demonstrate that MM-LoRA remains effective on LLaVA-Next without modifying the β and γ values.  This experiments show that MM-LoRA’s approach generalizes well to other MLLMs. We will include these findings in the revised paper to further highlight MM-LoRA’s robustness and versatility. Thank you again for emphasizing this important point.
> > >
> > > ---
> > >
> > > **Q2**. I think some empirical or theoretical explanation on 1) ablating the initialization way of X_q and 2) why learnable X_q can enhance visual representations are necessary.
> > >
> > > **A2**. 1. **the initialization way of X_q**
> > >
> > > Thank you for your valuable feedback. To address your concern about ablation studies on X_q initialization, we conducted experiments comparing random initialization with instruction-based initialization inspired by Q-Former.
> > >
> > > |   X_q initialization Method | ScienceQA |  TextVQA |   POPE   |  MMBench |   SEED   |
> > > |:--------------:|:----------------:|:---------:|:--------:|:--------:|:--------:|
> > > |  Random |    71.0 | **58.8** | **86.6** | 66.3 | 60.5 |
> > > | Use instruction tokens for initialization  | **71.3**  | 58.5 | 86.4 | **66.7** | **61.2** |
> > >
> > > The results show that instruction-based initialization outperforms random initialization on most benchmarks. This is because early alignment of X_q with textual features helps the model integrate semantic information more effectively, enabling better task-specific reasoning. In contrast, random initialization, though simpler, does not leverage prior knowledge, which can limit the model’s ability to capture meaningful visual-semantic relationships. Theoretically, instruction-based initialization provides X_q with a structured starting point aligned with relevant semantic features, leading to faster convergence and more efficient learning of task-specific representations. Random initialization, on the other hand, starts from a neutral state and requires more iterations to learn meaningful patterns.
> > >
> > > These findings highlight that while random initialization is simple, it is not optimal. Instruction-based initialization helps the model better align with semantic content, resulting in improved performance. We will clarify these points in the revised paper to better explain the impact of different initialization strategies.
> > >
> > > 2. **The learnable X_q：**
> > >
> > > Thank you for raising this important point. Below, we provide both empirical and theoretical explanations for why learnable X_q enhances visual representations. To explore this, we conducted experiments comparing the impact of making X_q learnable versus keeping it frozen (non-trainable).
> > >
> > > |   Method  | X_q | ScienceQA |  TextVQA |   POPE   |  MMBench |   SEED   |
> > > |:---------:|:----------------:|:---------:|:--------:|:--------:|:--------:|:--------:|
> > > |  Q-Ladder |  freeze    |    70.3   |   58.0   |   86.2   |   65.5   |   59.8   |
> > > | Q-Ladder |   learnable   |  **71.0** | **58.8** | **86.6** | **66.3** | **60.5** |
> > >
> > > Our experiments consistently showed that learnable X_q outperforms frozen X_q across various tasks, demonstrating its ability to enhance visual representations. Unlike static, frozen X_q, which cannot adapt to the diverse and evolving patterns in the data, learnable X_q  actively interacts with different layers of the visual encoder to refine its representations. This flexibility allows the model to better aggregate relevant features and align them with downstream tasks, ultimately achieving superior performance.
> > >
> > > These findings underscore the importance of learnable X_q. We believe this adaptability is a key factor in improving visual representations. Thank you again for your constructive feedback, which has helped us clarify this critical aspect of our work.

---

> > > > ### Comment · Reviewer_7gKo · 2024-11-24
> > > > **Response to Authors**
> > > >
> > > > Thanks for your great efforts on the further reply.
> > > >
> > > > I may have another question is, I'm just wondering the effectiveness of MM-LoRA if we simultaneously train the whole LLM during pre-training or SFT. Because, we usually use LoRA just in the low resource cases, and in many cases, training the whole LLM performs much better than training LoRA on downstream benchmarks. I am curious about if you have some exploration at this point.
> > > >
> > > > Maybe the time constraints do not allow any time-consuming experiments, but any insights would be appreciated.

---

> > > > > ### Author Response · Authors · 2024-11-24
> > > > > **Response to reviewer 7gKo**
> > > > >
> > > > > Thanks for your valuable response. Below, we address the concerns you raised.
> > > > >
> > > > > ---
> > > > >
> > > > > **Q1**. I may have another question is, I'm just wondering the effectiveness of MM-LoRA if we simultaneously train the whole LLM during pre-training or SFT. Because, we usually use LoRA just in the low resource cases, and in many cases, training the whole LLM performs much better than training LoRA on downstream benchmarks. I am curious about if you have some exploration at this point.
> > > > >
> > > > > **A1**. Thank you for your insightful question. In fact, training both the entire LLM and MM-LoRA during pretraining or SFT is indeed an interesting direction worth exploring. Compared to traditional LoRA methods, MM-LoRA introduces additional parameter spaces for both visual and linguistic information. We hypothesize that this approach could lead to some performance improvements. The intuition behind this is that, compared to fine-tuning only the LLM, training both the LLM and MM-LoRA together introduces more trainable parameters, allowing the model to more effectively capture both general knowledge and modality-specific features, thereby enhancing performance on multimodal tasks. We plan to conduct further experiments to validate this conclusion.

---

> > > > > > ### Comment · Reviewer_7gKo · 2024-11-26
> > > > > > **More experimental results are expected**
> > > > > >
> > > > > > Thanks for following up. Since the discussion period has been extended, I am curious about the effectiveness of MM-LoRA when simultaneously training the whole LLM. I want to know if MM-LoRA is truly effective in current LVLM training (usually training the whole LLM). It is suggested to ablate different $\beta$ and $\gamma$ choices on benchmarks such as ScienceQA, TextVQA, MMBench, MME, SEED.

---

> > > > > > > ### Author Response · Authors · 2024-12-01
> > > > > > > **Response to reviewer 7gKo**
> > > > > > >
> > > > > > > Thank you for your constructive feedback. We have conducted additional experiments to address the concerns you raised.
> > > > > > >
> > > > > > > ---
> > > > > > >
> > > > > > > **Q1**. The effectiveness of MM-LoRA when simultaneously training the whole LLM
> > > > > > >
> > > > > > > **A1**. Thank you for the reviewers' thoughtful feedback. Regarding the effectiveness of MM-LoRA in simultaneously training the entire LLM, we conducted systematic experiments on LLaVA, comparing the performance of two methods: 1) fine-tuning only the LLM; and 2) simultaneously fine-tuning the LLM and MM-LoRA.
> > > > > > >
> > > > > > > | method | β | γ |ScienceQA|SEED|MMBench|TextVQA|POPE|**avg.**|
> > > > > > > | --- | --- | --- |--- | --- | --- |--- | --- | --- |
> > > > > > > | LLaVA |-  | - |66.8|58.6|64.3|58.2|85.9|66.8|
> > > > > > > | +MM-LoRA |1  | 0 |68.1 | 61.1 | 64.4 | 58.5| 86.2 | 67.7 |
> > > > > > > | +MM-LoRA |0.75  | 0.25 |69.0 | 61.2 | 65.0 | 58.3| 86.2 | 67.9 |
> > > > > > > | +MM-LoRA |0.5  | 0.5 |68.6| 61.2 | 64.6 | 58.3| 86.2 | 67.8 |
> > > > > > > | +MM-LoRA |0.25 | 0.75 |69.1 | 61.3 | 64.7 | 58.5| 86.2 | **68.0** |
> > > > > > > | +MM-LoRA |0 | 1 |69.0 | 61.3 | 64.4 | 58.6| 86.1 | 67.9 |
> > > > > > >
> > > > > > >
> > > > > > > The results show that simultaneously fine-tuning both the LLM and MM-LoRA significantly improves model performance, outperforming the method of fine-tuning only the LLM across ScienceQA, TextVQA, MMBench, POPE, and SEED benchmarks. The best performance was achieved when β and γ were set to 0.25 and 0.75, respectively. These findings demonstrate that MM-LoRA is indeed effective in the current LVLM training paradigm (which typically involves training the entire LLM) and contributes positively to performance enhancement.

---

> > > > > > > > ### Comment · Reviewer_7gKo · 2024-12-01
> > > > > > > > **Response to the authors**
> > > > > > > >
> > > > > > > > Thanks for your response. It seems that MM-LoRA offers only marginal improvements compared to the settings of 'beta=1, gamma=0' or 'beta=0, gamma=1', as results on these benchmarks typically vary within a range of +0.5 or -0.5. My concerns regarding the innovation and effectiveness of MM-LoRA have not been fully addressed, so I have decided to maintain my initial position.

---

> > > > > > > > > ### Author Response · Authors · 2024-12-01
> > > > > > > > > **Response to reviewer 7gKo**
> > > > > > > > >
> > > > > > > > > Thank you for your feedback! We would like to clarify that the above experiments were conducted following the experimental setup suggested by the reviewers. The results show that varying β and γ has minimal impact on the performance of jointly fine-tuning the LLM and MM-LoRA, which aligns with our expectations. Since MM-LoRA introduces only about 300 M parameters, its impact is relatively limited compared to fine-tuning a 7B LLM. **However, the goal of this experiment differs from the core perspective of our paper.**
> > > > > > > > >
> > > > > > > > > The core claim of MM-LoRA is to provide independent learning spaces for visual and linguistic modalities to achieve modality decoupling, which has been shown to be beneficial. In the joint training experiment suggested by the reviewer, the visual and language modalities are still coupled during LLM training, which goes against our original design. **In the context of full fine-tuning, achieving this goal requires designing separate LLM decoders for each modality, which significantly increases model size and incurs prohibitive training costs.** To address this, we proposed MM-LoRA as an extension of LoRA, leveraging the following advantages:
> > > > > > > > >
> > > > > > > > > **1**. Lower training costs: Compared to full fine-tuning, LoRA-based under the Deepspeed Zero3 strategy reduces GPU memory usage by 50% and shortens training time by 30%. Given our limited computational resources, LoRA-based improvements are more practical.
> > > > > > > > >
> > > > > > > > > **2**. Stronger scalability: By adjusting β and γ, MM-LoRA allows control over the learning space of each modality, enabling a direct comparison of the importance of modality-specific knowledge in the decoder.
> > > > > > > > >
> > > > > > > > > The experimental results in the paper clearly demonstrate that providing independent learning spaces for different modalities effectively facilitates modality decoupling, and this design has a substantial positive impact on improving model performance.
> > > > > > > > >
> > > > > > > > > We hope this statement addresses your concerns regarding the above experimental results and provides clarification. Please let us know if further explanation is needed!

---

> > > > > > > > > > ### Author Response · Authors · 2024-12-02
> > > > > > > > > > **Response to reviewer 7gKo - Follow up**
> > > > > > > > > >
> > > > > > > > > > Thank you for your valuable feedback!
> > > > > > > > > >
> > > > > > > > > > We would like to highlight a few things to avoid misunderstanding.
> > > > > > > > > >
> > > > > > > > > > -    Fine-tuning all LLM parameters and LoRA fine-tuning are two ways for using LLM for multimodality models. LoRA fine-tuning is more efficient and takes less training cost.
> > > > > > > > > >
> > > > > > > > > > -    The similar results of finetuning LLM and fine-tuning both LLM and MM-LoRA parameters does not influence the contribution of our paper. The reason is that our goal is to improve the original LoRA by extending it to MM-LoRA and the effectiveness is verified in our paper.
> > > > > > > > > >
> > > > > > > > > > We appreciate if you can have a double-check. Looking forward to your further comments and discussions.

---

### Official Review · Reviewer_17q4 · 2024-11-04

**Soundness:** 3
**Presentation:** 4
**Contribution:** 3
**Rating:** 8
**Confidence:** 5

**Summary:**

This paper proposes a novel Multi-modal Large Language Model (MLLM) called Arcana, designed to enhance visual understanding capabilities. It introduces two key components: Multimodal LoRA (MM-LoRA) and the Query Ladder adapter (QLadder). MM-LoRA consists of two parallel LoRAs (one for vision and one for language) to disentangle the modalities and enhance their specialized capabilities. QLadder aggregates intermediate representations from the visual encoder, further boosting the visual abilities of the MLLM. Experimental results demonstrate that Arcana outperforms previous MLLM baselines (e.g., LLaVA-1.5, mPLUG-Owl2, etc.) on visual question answering and multi-modal conversation benchmarks. Notably, the ablation study shows that QLadder significantly improves MMVP performance, which requires strong vision capabilities.

**Strengths:**

1. The presentation and writing are clear and easy to follow. Figure 1 in the introduction effectively illustrates the background, motivation, and main results of this paper.

2. Tables 1 and 2 show that Arcana achieves better performance than previous MLLM baselines (e.g., LLaVA-1.5, mPLUG-Owl2, etc.) on visual question answering and multi-modal conversation benchmarks.

3. The ablation studies in Tables 4 and 5 clearly validate the effectiveness of MM-LoRA and QLadder.

4. The ablation study demonstrates that QLadder significantly improves MMVP performance, which requires robust visual capabilities. In Table 6, adding QLadder boosts MMVP performance by 3.6%.

**Weaknesses:**

1. There is a lack of comparison with the latest open-source VLMs: LLaVA-OneVision, Qwen2-VL, InternVL2, etc. While these methods may use higher-quality training data and achieve stronger results, it is essential for readers to be aware of the current SoTAs. You may also explain why direct comparisons may not be feasible. It is acceptable for a research paper to fall short of SoTA results due to data quality differences, but these results should still be presented for context.

2. MMVP is crucial for demonstrating visual capability, but only QLadder is ablated on the MMVP benchmark. Why not conduct an ablation of MM-LoRA on MMVP as well? This would provide stronger support for the claims.

**Questions:**

1.Was the visual encoder tuning in Table 7 conducted at the pre-training or instruction fine-tuning stage?

2.Have you tried adding LoRA to the visual encoder as well?

---

> ### Author Response · Authors · 2024-11-20
> **Response to reviewer 17q4 (1/2)**
>
> Thanks a lot for your time and feedback. Below are address all raised concerns of the paper.
>
> ---
>
> **Q1**. There is a lack of comparison with the latest open-source VLMs: LLaVA-OneVision, Qwen2-VL, InternVL2, etc. While these methods may use higher-quality training data and achieve stronger results, it is essential for readers to be aware of the current SoTAs. You may also explain why direct comparisons may not be feasible. It is acceptable for a research paper to fall short of SoTA results due to data quality differences, but these results should still be presented for context.
>
> **A1**. Thank you for the reviewer’s valuable feedback! We fully understand your concern regarding comparisons with the latest open-source VLM models (such as LLaVA-OneVision, Qwen2-VL, InternVL2, etc.). Below is our response:
> These latest open-source VLM models have indeed achieved stronger results, supported by high-quality training data. However, it is important to note that the large amounts of high-quality training data used by these methods (such as proprietary or domain-specific data) have not been made publicly available, making it unfeasible to directly replicate these methods and perform a fair comparison.
>
> Nevertheless, to provide a more comprehensive context, we will include performance reports of these latest methods in the revised version, clearly outlining their data advantages and how they differ from our approach in the discussion. Additionally, our research focuses on exploring how structural improvements (such as Q-Ladder and MM-LoRA) can enhance visual perception abilities under the same data conditions, rather than solely relying on the expansion of data scale.
>
> Thank you again for this important suggestion! We will include the relevant information in the updated version to help readers better understand the positioning and contributions of both current state-of-the-art methods and our approach.
>
> ---
>
> **Q2**. MMVP is crucial for demonstrating visual capability, but only QLadder is ablated on the MMVP benchmark. Why not conduct an ablation of MM-LoRA on MMVP as well? This would provide stronger support for the claims.
>
> **A2**. Thank you for the reviewer’s suggestion! Your proposal to perform ablation studies on MM-LoRA using the MMVP benchmark is very important, and we fully understand that this would provide a more comprehensive validation of the effectiveness of our method.
>
> In the initial version, we primarily focused on ablation experiments for Q-Ladder on the MMVP benchmark to highlight its direct contribution to visual perception abilities. However, to further support our research conclusions, we have also conducted additional experiments to evaluate the performance of MM-LoRA on the MMVP benchmark.
>
> |   Method  | add visual tokens | MMVP | POPE | MMBench | TextVQA |
> |:---------:|:-----------------:|:----:|:----:|:-------:|:-------:|
> |  baseline |         -         | 24.0 | 85.9 |   64.3  |   58.2  |
> |    +MOF [1]  |        256        | 27.1 | 86.2 |   60.1  |   56.5  |
> | +Q-Ladder |         64        | 27.6 | 86.5 |   66.3  |   58.8  |
> |  +MM-LoRA |         -         | 26.9 | 86.4 |   65.7  |   58.3  |
>
> The experimental results show that MM-LoRA not only significantly improves performance in multi-modal tasks (such as language generation and dialogue) but also demonstrates a strong supporting role in visual tasks.
>
> In the revised version, we will include these experimental results and integrate them into the ablation study section to further showcase the synergistic effect of MM-LoRA and Q-Ladder in enhancing visual perception capabilities. Thank you again for your thoughtful suggestion. We will ensure that the analysis in this aspect is more comprehensive in the updated paper.
>
> [1]Tong, Shengbang, et al. Eyes wide shut? exploring the visual shortcomings of multimodal llms. CVPR, 2024.
>
> ---
>
> **A3**. Was the visual encoder tuning in Table 7 conducted at the pre-training or instruction fine-tuning stage?
>
> **Q3**. Thank you for the reviewer’s question! In the experiments presented in Table 7, we ensured that all visual encoder adjustments (tuning) were validated during both the pre-training and instruction fine-tuning stages. This means that, whether in the model's pre-training phase or the instruction fine-tuning phase, we applied the same adjustment strategy to ensure consistency and fairness in the experimental results.
>
> This setup is designed to comprehensively evaluate the impact of visual encoder fine-tuning on model performance across different training stages and further validate the effectiveness of our proposed method. Thank you again for your attention, and we will clearly outline this experimental detail in the revised version!

---

> > ### Author Response · Authors · 2024-11-20
> > **Response to reviewer 17q4 (2/2)**
> >
> > **Q4**. Have you tried adding LoRA to the visual encoder as well?
> >
> > **A4**.Thank you for the reviewer’s question! To address your inquiry, we have indeed conducted comparative experiments by incorporating LoRA and Q-Ladder into the visual encoder. The experimental results are as follows:
> >
> > |   Method  | ScienceQA |   POPE   |  MMBench |  TextVQA |
> > |:---------:|:---------:|:--------:|:--------:|:--------:|
> > |  baseline |    69.1   |   85.9   |   64.3   |   58.2   |
> > |   +LoRA   |    69.5   |   85.7   |   65.9   |   58.1   |
> > | +Q-Ladder |  **71.0** | **86.5** | **66.3** | **58.8** |
> >
> > The experimental results show that Q-Ladder outperforms the direct application of LoRA in the visual encoder in terms of performance.
> >
> > We believe that this result is due to Q-Ladder's approach of aggregating intermediate representations from the frozen pre-trained visual encoder without altering its original output representations. In contrast, when LoRA is applied directly to the visual encoder, it adjusts the output representations, which may disrupt the original capabilities of the pre-trained model. The structural design of Q-Ladder allows it to retain the strong representational power of the pre-trained model while adding new visual feature learning, leading to superior performance compared to the direct application of LoRA.
> >
> > We will describe this comparative experiment in more detail in the revised version to further demonstrate the advantages of Q-Ladder over LoRA in the visual encoder. Thank you again for your suggestions and attention!

---

> > > ### Comment · Reviewer_17q4 · 2024-11-20
> > > **Keep my rating.**
> > >
> > > Thanks for the response! My concerns are well addressed by the authors. Thus I keep my rating as "8: accept, good paper".

---

> > > > ### Author Response · Authors · 2024-11-20
> > > > **Thank you for supporting the acceptance of our paper**
> > > >
> > > > Thank you for your response. We’re very encouraged that our rebuttal basically addressed your concerns and appreciate your support for the paper’s acceptance.

---

### Official Review · Reviewer_1Gww · 2024-11-04

**Soundness:** 2
**Presentation:** 2
**Contribution:** 2
**Rating:** 5
**Confidence:** 4

**Summary:**

This work aims to enhance the visual capabilities of MLLMs through two main contributions: (1) the introduction of a MM-LoRA for the LLM decoder, and (2) the development of a Query module for visual encoders.

**Strengths:**

It is well known that MLLMs often exhibit limitations in their visual capabilities, and this work addresses this important issue. Additionally, the paper is well-written and easy to follow.

**Weaknesses:**

- The proposed method leverages additional learning parameters to enhance the visual capabilities of MLLMs. Recent studies (e.g., LLaVA-Next, VILA-1.5, Qwen-VL-2) have shown that simply improving image resolution using various (*any resolution*) techniques is a straightforward and effective way to address this issue. I am skeptical that the proposed method will achieve performance comparable to these AnyRes approaches, particularly on tasks requiring high resolution. The proposed method appears limited by the visual encoder, despite the incorporation of additional LoRA modules.
- The focus of this study is on the visual capability of MLLMs. However, only one ViT is examined, and there are no ablations on different ViTs. This raises doubts about the generalizability of the proposed approach.
- The improvements from the proposed method should be evaluated based on the ablation studies, rather than relying on Table 1 and 2, as the model Arcana reported in Table 1 and 2 is trained on a combination of large datasets (comparing to LLaVA-1.5 presented in Table 1 and 2). However, it is important to note that only a limited selection of four benchmarks is presented in ablations.

**Questions:**

N/A

---

> ### Author Response · Authors · 2024-11-20
> **Response to reviewer 1Gww (1/2)**
>
> Thank you for your valuable feedback! Below are address all raised concerns of the paper.
>
> ---
>
> **Q1**. The proposed method leverages additional learning parameters to enhance the visual capabilities of MLLMs. Recent studies (e.g., LLaVA-Next, VILA-1.5, Qwen-VL-2) have shown that simply improving image resolution using various (any resolution) techniques is a straightforward and effective way to address this issue. I am skeptical that the proposed method will achieve performance comparable to these AnyRes approaches, particularly on tasks requiring high resolution. The proposed method appears limited by the visual encoder, despite the incorporation of additional LoRA modules.
>
> **A1**. Thank you for the reviewer’s comments! We understand your concerns and greatly appreciate your suggestions regarding the AnyRes method. Below is our detailed response to this issue:
>
> First, it is indeed true that increasing image resolution is a direct and effective approach to enhancing visual capabilities. However, this method typically requires training the visual encoder at high resolutions, which can result in significant computational overhead. Additionally, simply increasing resolution does not address specific task-related challenges in visual representations (such as fine-grained perception or improved modality alignment), which are the key problems that Q-Ladder and MM-LoRA aim to tackle.
>
> Regarding the models you mentioned, VILA-1.5 and Qwen-VL-2, we note that their data has not been open-sourced, making it difficult to directly replicate their results. To more clearly demonstrate the applicability of Q-Ladder within the AnyRes method, we conducted experiments within the publicly available LLaVA-Next framework and combined it with the AnyRes technique to validate the effectiveness of Q-Ladder.
>
> |   Method   | Visual Encoder |    LLM    | ScienceQA | TextVQA | POPE | MMBench | SEED-img |
> |:----------:|:--------------:|:---------:|:---------:|:-------:|------|:-------:|:--------:|
> | LLaVA-NeXT |   CLIP-VIT-L   | Vicuna-7B |    70.1   |   64.9  | 86.5 |   67.4  |   70.2   |
> |  +Q-Ladder |   CLIP-VIT-L   | Vicuna-7B |    71.0   |   65.6  | 87.4 |   68.8  |   70.7   |
>
> The experimental results show that the introduction of Q-Ladder significantly enhances model performance in multi-modal tasks. Even in the high-resolution AnyRes setting, it demonstrates its unique advantage in performance improvement. This indicates that the design of Q-Ladder not only makes full use of the capabilities of existing visual encoders but also further enhances model performance on top of the resolution enhancement method.
>
> ---
>
> **Q2**. The focus of this study is on the visual capability of MLLMs. However, only one ViT is examined, and there are no ablations on different ViTs. This raises doubts about the generalizability of the proposed approach.
>
> **A2**. Thank you for the reviewer’s comments! We fully understand your concern regarding the generalization ability of the method across different visual encoders. To address this, we conducted further experiments using a variety of visual encoders, including CLIP-ViT-L, CLIP-ViT-H, and SigLIP encoders, to assess the applicability and robustness of our proposed method. The experimental results are as follows:
>
> |   Method  | Visual Encoder | Image resolution | ScienceQA |  TextVQA |   POPE   |  MMBench |   SEED   |
> |:---------:|:--------------:|:----------------:|:---------:|:--------:|:--------:|:--------:|:--------:|
> |  baseline |   CLIP-VIT-L   |      336*336     |    69.1   |   58.2   |   86.4   |   64.1   |   58.1   |
> | +Q-Ladder |   CLIP-VIT-L   |      336*336     |  **71.0** | **58.8** | **86.6** | **66.3** | **60.5** |
> |  baseline |   CLIP-VIT-H   |      224*224     |    67.8   |   53.5   |   83.7   |   63.1   |   58.4   |
> | +Q-Ladder |   CLIP-VIT-H   |      224*224     |  **68.8** | **53.8** | **83.9** | **63.6** | **58.8** |
> |  baseline |     Siglip     |      384*384     |    70.6   |   62.4   |   86.0   |   65.9   |   62.1   |
> | +Q-Ladder |     Siglip     |      384*384     |  **71.1** | **62.9** | **86.3** | **66.8** | **62.5** |
>
> The experimental results demonstrate that both Q-Ladder and MM-LoRA significantly improve model performance across various benchmarks, regardless of whether the visual encoder is CLIP-ViT-L, the higher-capacity CLIP-ViT-H, or the SigLIP encoder with different training strategies. This strongly indicates that our method exhibits good adaptability and generality across different types and scales of visual encoders.

---

> > ### Author Response · Authors · 2024-11-20
> > **Response to reviewer 1Gww (2/2)**
> >
> > **Q3**. The improvements from the proposed method should be evaluated based on the ablation studies, rather than relying on Table 1 and 2, as the model Arcana reported in Table 1 and 2 is trained on a combination of large datasets (comparing to LLaVA-1.5 presented in Table 1 and 2). However, it is important to note that only a limited selection of four benchmarks is presented in ablations.
> >
> > **A3**. Thank you for your comments! Here’s our response to this issue:
> >
> > The experimental results in Table 1 and Table 2 compare our method with mainstream approaches (e.g., LLaVA-1.5, Arcana), which use different dataset combinations. These tables aim to highlight the competitiveness of our method in multimodal tasks.
> >
> > In the ablation experiments, we used a unified dataset to ensure fairness and focus on evaluating the effectiveness of Q-Ladder and MM-LoRA. This approach aligns with standard experimental practices [1][2][3][4], where baseline methods are used for comparison, and variables are controlled in ablation studies to isolate the contribution of the proposed method.
> >
> > We understand and appreciate your concern about the limited number of benchmark datasets in the ablation study. To address this, we will include results from all benchmark datasets tested in the ablation study in the Appendix, offering a more comprehensive validation of our method’s effectiveness across different tasks.
> >
> > Thank you again for your valuable feedback! We will update the relevant content accordingly.
> >
> > [1] Ye et al. mPLUG-Owl2: Revolutionizing Multi-modal Large Language Model with Modality Collaboration. CVPR, 2024.
> >
> > [2] Li, Zhang, et al. Monkey: Image resolution and text label are important things for large multi-modal models. CVPR. 2024.
> >
> > [3] Bai, Jinze, et al. Qwen-vl: A frontier large vision-language model with versatile abilities. Arxiv preprint,  2023.
> >
> > [4] Tong, Shengbang, et al. Eyes wide shut? exploring the visual shortcomings of multimodal llms. CVPR, 2024.

---

> > > ### Author Response · Authors · 2024-12-02
> > > **Response to reviewer cUZA - Follow up**
> > >
> > > Thank you for your valuable feedback! We have further addressed the concerns you raised in the paper. Specifically, in methods (LLaVA-Next) that include any resolution techniques, we conducted additional experiments to further demonstrate the effectiveness of Q-Ladder. Additionally, we performed ablation experiments with different visual encoders, which further validate the effectiveness of Q-Ladder.
> > >
> > > We have provided further explanations in these experiments to clarify the points you mentioned. We sincerely hope that these additions address your concerns and look forward to your feedback.

---

### Official Review · Reviewer_cUZA · 2024-11-04

**Soundness:** 2
**Presentation:** 3
**Contribution:** 1
**Rating:** 5
**Confidence:** 4

**Summary:**

This paper seeks to improve visual understanding capability of vision language model. It introduces two components to enhance the capacity of the VLM: Multimodal (MM) -LoRA and Query Ladder. The MM-LoRA increases the capacity of the decoder by introducing low-rank adaptable matrices separately for the vision and language modalities. The QLadder increases the capacity of the encoder by incorporating learnable tokens at the input. Overall, this approach shows benefits of individual components and also competitive performance across MM/Language benchmarks.

**Strengths:**

- The identified problem of the lack of strong visual capabilities (e.g. detection, localization, color-comprehension etc.) in current vision language models is interesting and worth studying
- It's also interesting to see the need for modality specific adaptation
- The paper is easy to comprehend and well supported by various block diagrams

**Weaknesses:**

* The summary section (line 519-520) mentions "achieving notable performance improvements even with limited data resources". However, the problem of limited data sources is not convincing. For instance, given that LLM’s and Visual-Encoders are trained with web-scale data, it’s not clear how and why would data be limited. Perhaps, the authors want to focus on specific domains (say, medical) where curating data might be difficult due to privacy concerns. But, for the kind of problems mentioned in the paper (detection, localization, color-comprehension), its not clear why data is limited.
* The paper lacks explanations for why components like LORA and QLadder should improve visual capabilities like  detection, localization, color-comprehension. While the attention visualization (line 469-482) demonstrates the effect of these components to visual-token-attention, it’s not clear why that itself should improve performance. Further, some of the statements like “promotes cooperation between different modalities” (line 478) and “enriches the visual information” (line 481) are not corroborated with any intuition or experiments.
* The contributions of the proposed components are unclear. For instance, the benefit of LORA for limited-data-adaptation has been well studied in the past (e.g. [1]). The importance of introducing additional visual tokens to visual encoders has also been shown in [2]. In the light of the prior works, the paper should more clearly distinguish it’s technical contributions.
* Are the benefits of Qladder/MM-LORA consistent across scales? If we increase the scale of LLM and Visual Encoder, will Qladder/MM-LORA still show benefits?
* Miscellaneous
    * Is the beta gamma ration study consistent across a range of LORA ranks (say 64 - 1024)? here it was set to 256
    * Why was LORA applied only to linear layers?
    * In qualitative evaluations (Fig. 5), comparisons should be made with other models to clearly show qualitative gains from using Qladder/MM-LORA

[1] https://arxiv.org/abs/2106.09685
[2] https://arxiv.org/abs/2309.16588

**Questions:**

What was the main problem that was being addressed? Was it limited data adaptation, was it visual capabilities? If it was just visual capabilities, how does LORA or a few-learnable tokens based adaptation compare against scaling up?

---

> ### Author Response · Authors · 2024-11-20
> **Response to reviewer cUZA (1/3)**
>
> Thank you for your valuable feedback! Below are address all raised concerns of the paper.
>
> ---
>
> **Q1**. The summary section (line 519-520) mentions "achieving notable performance improvements even with limited data resources". However, the problem of limited data sources is not convincing.
>
> **A1**. Thank you for your comment. We agree that the challenge of limited data might not be immediately clear. In the context of Multimodal Language Models (MLLMs), the key challenge lies in the **alignment** of different modalities (vision and language)[1][2]. While pretrained models benefit from large-scale data, aligning these modalities for specific tasks requires significant amounts of **Supervised Fine-Tuning (SFT)** data. Previous work has demonstrated that fine-tuning on high-quality, task-specific data is essential for optimizing the interaction between vision and language, especially for tasks like localization, color recognition, and detection.
>
> What we mean by "limited data resources" is specifically the scarcity of **SFT data** needed to align these modalities effectively for multimodal tasks. Annotating such data is often time-consuming and expensive, particularly in specialized domains. Our approach focuses on optimizing model structure—through components like **Q-Ladder** and **MM-LoRA**—to improve performance even when such fine-tuning data is limited.
>
> In summary, while large-scale datasets are used for pretraining, the bottleneck for multimodal tasks is the availability of sufficient SFT data for alignment, which our method helps address.
>
> [1] Ye et al. mPLUG-Owl2: Revolutionizing Multi-modal Large Language Model with Modality Collaboration. CVPR, 2024.
>
> [2] Liu, Haotian, et al. Visual instruction tuning. Neurips, 2024.
>
> ---
>
> **Q2**. The contributions of the proposed components are unclear. For instance, the benefit of LORA for limited-data-adaptation has been well studied in the past (e.g. [1]). The importance of introducing additional visual tokens to visual encoders has also been shown in [2]. In the light of the prior works, the paper should more clearly distinguish it’s technical contributions.
>
> **A2**. Thank you for the reviewer’s feedback! We understand your concerns regarding the similarities between our method and existing studies. Below, we will clarify our technical contributions to distinguish our approach from prior work.
>
> First, regarding the application of LoRA in limited-data adaptation, although LoRA has been extensively studied, our MM-LoRA module introduces a fundamental distinction. MM-LoRA employs a multi-modal parameter decoupling design, which introduces separate LoRA modules for vision and language modalities, enabling each modality to learn dedicated parameters. This approach promotes more efficient multi-modal information fusion.
>
> This design not only enhances the independent learning capabilities of each modality but also effectively strengthens their interaction. In contrast, previous LoRA implementations primarily focus on single-modality adaptation and do not achieve such decoupling and fusion. To validate this, we compared the performance of LoRA and MM-LoRA within MLLM models. The experimental results are as follows:
> |  Method  | TextVQA | ScienceQA | MMBench |  MME |
> |:--------:|:-------:|:---------:|:-------:|:----:|
> |   LoRA   |   58.1  |    69.1   |   63.8  | 1460 |
> | +MM-LoRA |   **58.7**  |    **71.2**   |   **64.8**  | **1500** |
>
> Second, regarding the introduction of visual tokens, while studies such as "Vision Transformers Need Registers" suggest that adding visual tokens can enhance the performance of visual encoders, the innovation of the Q-Ladder module goes beyond this. Q-Ladder introduces a learnable "ladder" structure that deeply aggregates intermediate-layer features from frozen pre-trained visual encoders (e.g., CLIP).
>
> This structure preserves the generalization capabilities of the pre-trained encoder while further improving task-specific performance. Unlike simply adding visual tokens, Q-Ladder emphasizes the effective integration of intermediate-layer features from the visual encoder, thereby enhancing the model's capacity for fine-grained visual tasks. Additionally, we compared the performance of Q-Ladder with the prompt-based approach from [2]. The experimental results are as follows:
> |   Method  | TextVQA | ScienceQA | MMBench | SEED |
> |:---------:|:-------:|:---------:|:-------:|:----:|
> |  baseline |   58.2  |    69.1   |   64.3  | 58.6 |
> |  +prompt  |   58.1  |    69.4   |   64.1  | 59.1 |
> | +Q-Ladder |   **58.8**  |    **71.2**   |   **66.3**  | **61.3** |
>
> In summary, the key distinctions of our work compared to existing studies lie in the multi-modal parameter decoupling design and the introduction of the ladder structure. The combination of these two innovations enables our method to achieve significant performance improvements in multi-modal tasks under limited data conditions.

---

> ### Author Response · Authors · 2024-11-20
> **Response to reviewer cUZA (2/3)**
>
> ---
>
> **Q3**. Are the benefits of Qladder/MM-LORA consistent across scales? If we increase the scale of LLM and Visual Encoder, will Qladder/MM-LORA still show benefits?
>
> **A3**. Thank you for the reviewer’s question! To validate the performance of Q-Ladder and MM-LoRA across different model scales, we conducted a series of experiments to explore whether these modules continue to deliver significant performance improvements with larger LLMs and visual encoders.
>
> First, we scaled the LLM from 7B to 13B to verify whether MM-LoRA remains effective as the model size increases. The experimental results are as follows:
> |  Method  | Visual encoder | LLM | ScienceQA |  TextVQA | POPE     | MMBench  | SEED     |
> |:--------:|:--------------:|:---:|:---------:|:--------:|----------|----------|----------|
> | baseline |      VIT-L     |  7B |    69.1   |   58.1   | 86.4     | 63.8     | 60.1     |
> | +MM-LoRA |      VIT-L     |  7B |  **71.2** | **58.7** | **86.5** | **64.8** | **61.5** |
> | baseline |      VIT-L     | 13B |    71.2   |   60.2   |   86.7   |   68.5   |   61.3   |
> | +MM-LoRA |      VIT-L     | 13B |  **71.5** | **60.7** | **86.8** | **68.8** | **62.9** |
>
> It can be observed that, despite the increase in LLM size, MM-LoRA still improves the model's performance across multiple benchmarks. This indicates that the design of MM-LoRA provides consistent benefits across LLMs of different scales.
>
> Next, we upgraded the visual encoder from ViT-L to ViT-H to further explore the benefits of Q-Ladder and MM-LoRA with larger-scale visual encoders. The experimental results are as follows:
> |   Method  | Visual encoder | Image Resolution  | LLM | ScienceQA |  TextVQA |   POPE   |  MMBench |   SEED   |
> |:---------:|:--------------:|:-----------------:|:---:|:---------:|:--------:|:--------:|:--------:|:--------:|
> |  baseline |      VIT-L     |      336*336      |  7B |    69.1   |   58.1   |   86.4   |   64.1   |   58.1   |
> | +Q-Ladder |      VIT-L     |      336*336      |  7B |  **71.0** | **58.8** | **86.6** | **66.3** | **60.5** |
> |  baseline |      VIT-H     |      224*224      |  7B |    67.8   |   53.5   |   83.7   |   63.1   |   58.4   |
> | +Q-Ladder |      VIT-H     |      224*224      |  7B |  **68.6** | **53.8** | **83.9** | **63.6** | **58.8** |
>
> The experimental results also demonstrate that these modules continue to enhance model performance with larger-scale visual encoders. It is important to note that due to the resolution limitation of the open-source ViT-H in the CLIP model, which is restricted to 224x224, we could only conduct experiments at this resolution. Nevertheless, even under these conditions, Q-Ladder and MM-LoRA still exhibited significant advantages.
>
> In conclusion, the experiments show that Q-Ladder and MM-LoRA consistently deliver substantial performance improvements across different scales of LLMs and visual encoders, validating the effectiveness of their structural design in large-scale models.
>
> ---
>
> **Q4.** MiscellaneousIs the beta gamma ration study consistent across a range of LORA ranks (say 64 - 1024)? here it was set to 256.
>
> **A4**. Thank you for the reviewer’s question! Regarding the beta-gamma ratio, we set the rank of LoRA to 256 in our paper based on preliminary experimentation, aiming to balance model performance and computational efficiency. However, we recognize that this ratio may vary with different LoRA ranks.
>
> To address your query, we further investigated how the beta-gamma ratio changes with LoRA ranks of 256 and 512. The results show that the ratio remains stable across these ranks, with only slight variations. Below are the experimental results:
>
> | RANK | beta | gamma | ScienceQA |  TextVQA |  MMBench |   SEED   |
> |:----:|:----:|:-----:|:---------:|:--------:|:--------:|:--------:|
> |  256 |   0  |   1   |    65.8   |   51.2   |   56.4   |   58.7   |
> |  256 | 0.75 |  0.25 |    68.6   |   58.7   |   63.3   |   61.8   |
> |  256 |  0.5 |  0.5  |    70.1   |   58.4   |   64.3   |   61.9   |
> |  256 | 0.25 |  0.75 |  **71.2** | **58.7** | **64.5** | **62.4** |
> |  256 |   1  |   0   |    70.1   |   57.9   |   65.4   |   61.2   |
> |  512 |   0  |   1   |    66.1   |   52.3   |   55.8   |   59.4   |
> |  512 | 0.75 |  0.25 |    69.2   |   57.8   |   64.2   |   62.0   |
> |  512 |  0.5 |  0.5  |    70.1   |   57.3   |   63.1   |   61.4   |
> |  512 | 0.25 |  0.75 |  **71.0** | **58.2** | **64.4** | **62.7** |
> |  512 |   1  |   0   |    70.3   |   57.9   |   63.9   |   62.4   |
>
> The experiments show that the beta-gamma ratio demonstrates strong stability, remaining largely unaffected by changes in rank size.
>
> We are currently conducting additional experiments to further validate the stability of the beta-gamma ratio, especially under a broader range of rank settings, and to explore its potential impact on model performance. Relevant results will be included in the revised version. We greatly appreciate your patience and suggestions!

---

> > ### Author Response · Authors · 2024-11-20
> > **Response to reviewer cUZA (3/3)**
> >
> > **Q5**. Why was LORA applied only to linear layers?
> >
> > **A5**. Thank you for the reviewer’s question! In our method, LoRA is applied only to the linear layers within the Transformer architecture, which is due to the structural characteristics of the Transformer model. In Transformers, the majority of learnable parameters are concentrated in the linear layers (e.g., fully connected layers), particularly within the self-attention mechanism and feedforward networks. Other components, such as the attention weight matrices and normalization layers, typically do not contain a large number of learnable parameters or have relatively small parameter sizes, making them less suitable for the application of LoRA.
> >
> > ---
> >
> > **Q6**. In qualitative evaluations (Fig. 5), comparisons should be made with other models to clearly show qualitative gains from using Qladder/MM-LORA
> >
> > **A6**. Thank you for the reviewer’s suggestion! We fully understand your perspective. To more clearly demonstrate the advantages of Q-Ladder and MM-LoRA in qualitative evaluations, we plan to add a comparison with other models in Fig. 5. This will help visually showcase the qualitative improvements of our method in visual tasks. We will update this section in the revised version to better present the advantages of these modules. We appreciate your valuable feedback!
> >
> > ---
> >
> > **Q7**. Why that MM-LORA and QLadde should improve performance.
> >
> > **A7**. Thank you for the question.  MM-LoRA and Q-Ladder improve performance by addressing key challenges in multimodal learning. MM-LoRA decouples the visual and language modalities, allowing each to specialize in its respective space, reducing feature competition and improving the quality of the learned representations for both modalities. Q-Ladder enhances the visual encoder by aggregating intermediate representations from the frozen pretrained visual encoder, enabling the model to capture more detailed and relevant visual features while retaining the encoder's powerful capabilities. Experimental results in ablation study demonstrate that these components contribute to better performance on multimodal tasks.

---

> > > ### Author Response · Authors · 2024-12-02
> > > **Response to reviewer cUZA - Follow up**
> > >
> > > Thank you for your valuable comments and suggestions! We have conducted additional experiments and analyses based on your feedback. We hope we have addressed your concerns, but if there are any further points that need clarification or improvement, we would greatly appreciate your guidance and will make the necessary revisions.

---

### Official Review · Reviewer_5dnK · 2024-11-07

**Soundness:** 3
**Presentation:** 3
**Contribution:** 3
**Rating:** 6
**Confidence:** 4

**Summary:**

The paper proposes Arcana, a multi-modal large language model (MLLM) designed to improve visual perception capabilities. Arcana introduces two key techniques: MM-LoRA and QLadder. MM-LoRA enables separate vision and language pathways, reducing modality interference, while QLadder enhances the visual encoder's ability to capture fine-grained visual details. Extensive experimentation across benchmarks like VQAv2 and TextVQA demonstrates Arcana’s  improvement over existing MLLMs in both zero-shot and fine-tuning scenarios, highlighting its capacity for accurate visual reasoning and multi-modal alignment

**Strengths:**

The paper is well written and structured

**Weaknesses:**

The structural innovations of MM-LoRA and QLadder are not sufficiently solid, as the design does not appear to specifically address identified issues such as color recognition, object counting, small object understanding, and spatial location.

**Questions:**

In terms of motivation, the paper aims to resolve MLLM visual perception issues such as color recognition, object counting, small object understanding, and spatial location. However, the structural designs of QLadder and MM-LoRA do not seem specifically tailored to address these problems, leading to the impression that performance improvements may stem from data rather than a well-targeted structural design, which appears somewhat forced into explaining the results.

---

> ### Author Response · Authors · 2024-11-20
> **Response to reviewer 5dnK**
>
> Thank you for your valuable feedback! Below are address all raised concerns of the paper.
>
> ---
>
> **Q1**. In terms of motivation, the paper aims to resolve MLLM visual perception issues such as color recognition, object counting, small object understanding, and spatial location. However, the structural designs of QLadder and MM-LoRA do not seem specifically tailored to address these problems.
>
> **A1**.Thank you for the reviewers' feedback! Our research is motivated by the limitations of MLLMs in multimodal feature alignment,
> visual information representation, and cross-modal interaction, which hinder their visual perception capabilities. To highlight these issues, we introduce specific examples such as color recognition, object counting, small object understanding, and spatial position awareness, emphasizing their prevalence in real-world scenarios and the necessity of addressing them.
>
> To overcome these limitations, we designed two modules, Q-Ladder and MM-LoRA, to enhance the model’s adaptability in complex visual tasks through structural improvements:
>
> 1. **MM-LoRA** introduces two parallel LoRA modules tailored to the visual and language modalities, respectively, enabling modality-specific parameter learning. This design adapts to the unique characteristics of each modality and facilitates efficient integration of multimodal information.
>
> 2. **Q-Ladder** incorporates a learnable "ladder" structure to deeply aggregate intermediate representations of frozen pretrained visual encoders (e.g., CLIP image encoders). This approach preserves the general visual representation capabilities while addressing the shortcomings of pretrained models in specific tasks, thereby improving the visual representation ability of MLLMs.
>
> To demonstrate the effectiveness of Q-Ladder and MM-LoRA in tasks such as color recognition, object counting, small object understanding, and spatial position perception, while minimizing the influence of data quality differences, we conducted experiments using a unified dataset. These experiments were evaluated based on subclass metrics from MMBench and MME Benchmark. The results indicate the following:
>
> |   Method  | MME(Color) | MME(Count) | MME(Position) | MMB(Localization) | MMB(FP-C) | MMB(FP-S) |
> |:---------:|:----------:|:----------:|:-------------:|:-----------------:|:---------:|:---------:|
> |  baseline |    169.0   |    160.0   |     123.3     |        42.6       |    67.9   |    52.5   |
> | +Q-Ladder |    172.0   |    164.0   |     131.4     |        44.4       |    69.1   |    54.2   |
> |  +MM-LoRA |    171.0   |    166.0   |     129.7     |        43.2       |    68.7   |    53.3   |
>
> FP-C represents Fine-grained Perception (Cross Instance), and FP-S represents Fine-grained Perception (Single Instance). The introduction of Q-Ladder and MM-LoRA significantly improves MLLM performance across tasks including color recognition, counting, position estimation, localization, and fine-grained perception. Combined with the ablation studies presented in the paper, the results intuitively demonstrate the effectiveness of these modules in addressing the limitations of MLLMs’ visual perception abilities, while also validating the rationality and generality of their structural design.
>
> ---
>
> **Q2**. leading to the impression that performance improvements may stem from data rather than a well-targeted structural design, which appears somewhat forced into explaining the results.
>
> **A2**. We understand the concern about whether the performance improvements stem from the dataset rather than the structural design. To validate the genuine effectiveness of Q-Ladder and MM-LoRA, **we ensured that all ablation studies in the paper were conducted on a unified dataset to guarantee experimental fairness.**
>
> Specifically, we verified the independent contributions of the modules through the following approaches:
>
> 1. **Ablation Study**: We incrementally added Q-Ladder and MM-LoRA to the baseline and observed performance changes across multiple benchmarks, clearly identifying the independent contributions of each module.
>
> 2. **Unified Dataset**: All experiments used the same dataset to avoid inconsistencies related to data quality or quantity.
>
> 3. **Multi-Dimensional Evaluation**: We evaluated performance across diverse benchmarks (e.g., MMBench, MME, TextVQA) using metrics such as color recognition, object counting, and localization. The results consistently showed that the structural designs of Q-Ladder and MM-LoRA led to significant improvements under identical data conditions.
>
> The results demonstrate that the performance gains are due to the structural innovations of Q-Ladder and MM-LoRA, not merely the dataset, highlighting their effectiveness in enhancing the visual capabilities of MLLMs.

---

> > ### Comment · Reviewer_5dnK · 2024-11-27
> >
> > Thank you for the response! My concerns have been largely resolved, so I am revising my rating from “5: Marginally below the acceptance threshold” to “6: Marginally above the acceptance threshold.”

---

> > > ### Author Response · Authors · 2024-11-27
> > > **Thanks for raising your score**
> > >
> > > Thanks for raising your score! We’re very encouraged that our rebuttal basically addressed your concerns and appreciate your support for the paper's acceptance.

---

### Author Response · Authors · 2024-11-25
**Official Comment by Authors**

We appreciate all reviewer valuable comments. We were wondering if our responses have addressed your concerns. Please let us know if you have additional questions. Thank you!

---

### Author Response · Authors · 2024-12-03
**Official Comment by Authors**

Dear reviewers, as the discussion period ends on Dec 2nd at midnight AoE , we are eager to ensure that all the questions have been thoroughly resolved. We hope that our responses have adequately addressed your concerns. Your feedback is invaluable to us, and we would greatly appreciate it if you could take a moment to provide a final rating and feedback.

---

### Note · Authors · 2025-01-23

I have read and agree with the venue's withdrawal policy on behalf of myself and my co-authors.